# Contrastive Corpus Attribution for Explaining Representations

**Chris Lin**∗, **Hugh Chen**∗, **Chanwoo Kim, Su-In Lee**
Paul G. Allen School of Computer Science & Engineering
University of Washington
Seattle, WA 98195, USA
{clin25,hughchen,chanwkim,suinlee}@cs.washington.edu

## Abstract

Despite the widespread use of unsupervised models, very few methods are designed to explain them. Most explanation methods explain a scalar model output. However, unsupervised models output representation vectors, the elements of which are not good candidates to explain because they lack semantic meaning. To bridge this gap, recent works defined a scalar explanation output: a dot product-based similarity in the representation space to the sample being explained (i.e., an explicand). Although this enabled explanations of unsupervised models, the interpretation of this approach can still be opaque because similarity to the explicand's representation may not be meaningful to humans. To address this, we propose *contrastive corpus similarity*, a novel and semantically meaningful scalar explanation output based on a reference *corpus* and a contrasting *foil* set of samples. We demonstrate that contrastive corpus similarity is compatible with many post-hoc feature attribution methods to generate *COntrastive COrpus Attributions (COCOA)* and quantitatively verify that features important to the corpus are identified. We showcase the utility of COCOA in two ways: (i) we draw insights by explaining augmentations of the same image in a contrastive learning setting (SimCLR); and (ii) we perform zero-shot object localization by explaining the similarity of image representations to jointly learned text representations (CLIP).

## 1 Introduction

Machine learning models based on deep neural networks are increasingly used in a diverse set of tasks including chess (Silver et al., 2018), protein folding (Jumper et al., 2021), and language translation (Jean et al., 2014). The majority of neural networks have many parameters, which impede humans from understanding them (Lipton, 2018). To address this, many tools have been developed to understand supervised models in terms of their prediction (Lundberg & Lee, 2017; Wachter et al., 2017). In this supervised setting, the model maps features to labels ($f : \mathcal{X} \rightarrow \mathcal{Y}$), and explanations aim to understand the model's prediction of a label of interest. These explanations are interpretable, because the label of interest (e.g., mortality, an image class) is meaningful to humans (Figure 1a).

In contrast, models trained in unsupervised settings map features to representations ($f : \mathcal{X} \rightarrow \mathcal{H}$). Existing supervised explanation methods can be applied to understand an individual element ($h_i$) in the representation space, but such explanations are not useful to humans unless $h_i$ has a natural semantic meaning. Unfortunately, the meaning of individual elements in the representation space is unknown in general. One possible solution is to enforce representations to have semantic meaning as in Koh et al. (2020), but this approach requires concept labels for every single training sample, which is typically impractical. Another solution is to enforce learned representations to be disentangled as in Tran et al. (2017) and then manually identify semantically meaningful elements to explain, but this approach is not post-hoc and requires potentially undesirable modifications to the training process.

**Related work.** Rather than explain a single element in the representation, approaches based on explaining the representation as a whole have recently been proposed, including RELAX (Wickstrøm et al., 2021) and label-free feature importance (Crabbé & van der Schaar, 2022) (Figure 1b)

---

∗Equal contribution.

(additional related work in Appendix A). These approaches both aim to identify features in the explicand (the sample to explain) that, when removed, point the altered representation away from the explicand's original representation.

Although RELAX and label-free feature importance successfully extend existing explanation techniques to an unsupervised setting, they have two major limitations. First, they only consider similarity to the explicand's representation; however, there are a variety of other meaningful questions that can be asked by examining similarity to other samples' representations. Examples include asking, "Why is my explicand similar to dog images?" or "How is my rotation augmented image similar to my original image?". Second, RELAX and label-free importance find features which increase similarity to the explicand in representation space from any direction, but in practice some of these directions may not be meaningful. Instead, just as human perception often explains by comparing against a contrastive counterpart (i.e., *foil*) (Kahneman & Miller, 1986; Lipton, 1990; Miller, 2019), we may wish to find features that move toward the explicand relative to an explicit "foil". As an example, RELAX and label-free importance may identify features which increase similarity to a dog explicand image relative to other dog images or even cat images; however, they may also identify features which increase similarity relative to noise in the representation space corresponding to unmeaningful out-of-distribution samples. In contrast, we can use foil samples to ask specific questions such as, "What features increase similarity to my explicand relative to cat images?".

**Contribution.** (1) To address the limitations of prior works on explaining unsupervised models, we introduce *COntrastive COrpus Attribution (COCOA)*, which allows users to choose *corpus* and *foil* samples in order to ask, "What features make my explicand's representation similar to my corpus, but dissimilar to my foil?" (Figure 1c). (2) We apply COCOA to representations learned by a self-supervised contrastive learning model and observe class-preserving features in image augmentations. (3) We perform object localization by explaining a mixed modality model with COCOA.

**Motivation.** Unsupervised models are prevalent and can learn effective representations for downstream classification tasks. Notable examples include contrastive learning (Chen et al., 2020) and self-supervised learning (Grill et al., 2020). Despite their widespread use and applicability, unsupervised models are largely opaque. Explaining them can help researchers understand and therefore better develop and compare representation learning methods (Wickstrøm et al., 2021; Crabbé & van der Schaar, 2022). In deployment, explanations can help users better monitor and debug these models (Bhatt et al., 2020).

Moreover, COCOA is beneficial even in a supervised setting. Existing feature attribution methods only explain the classes the model has been trained to predict, so they can only explain classes which are fully labeled in the training set. Instead, COCOA only requires a few class labels after the training process is complete, so it can be used more flexibly. For instance, if we train a supervised model on CIFAR-10 (Krizhevsky et al., 2009), existing methods can only explain the ten classes the model was trained on. Instead, we can collect new samples from an unseen class, and apply COCOA to a representation layer of the trained model to understand this new class.

## 2 NOTATION

We consider an arbitrary input space $\mathcal{X}$ and output space $\mathcal{Z}$. Given a model $f : \mathcal{X} \to \mathcal{Z}$ and an explicand $\boldsymbol{x}^e \in \mathcal{X}$ to be explained, local feature attribution methods assign a score to each input feature based on the feature's importance to a scalar *explanation target*. A model's intermediate or final output is usually not a scalar. Hence, feature attribution methods require an *explanation target function* $\gamma_{f,*} : \mathcal{X} \to \mathbb{R}$ that transforms a model's behavior into an explanation target. The subscript $f$ indicates that a model is considered a fixed parameter of an explanation target function, and $*$ denotes an arbitrary number of additional parameters. To make this concrete, consider the following example with a classifier $f^{class} : \mathcal{X} \to [0, 1]^C$, where $C$ is the number of classes.

**Example 2.1.** *Given an explicand $\boldsymbol{x}^e \in \mathcal{X}$, let the explanation target be the predicted probability of the explicand's predicted class. Then the explanation target function is*

$$\gamma_{f^{class}, \boldsymbol{x}^e}(\boldsymbol{x}) = f^{class}_{\arg\max_{j=1,\ldots,C} f^{class}_j(\boldsymbol{x}^e)}(\boldsymbol{x}),$$

*for all $\boldsymbol{x} \in \mathcal{X}$, where $f^{class}_i(\cdot)$ denotes the $i$th element of $f^{class}(\cdot)$. Here, the explanation target function has the additional subscript $\boldsymbol{x}^e$ to indicate that the explicand is a fixed parameter.*

Let $\mathbb{R}^{\mathcal{X}}$ denote the set of functions that map from $\mathcal{X}$ to $\mathbb{R}$. Formally, a local feature attribution method $\phi : \mathbb{R}^{\mathcal{X}} \times \mathcal{X} \to \mathbb{R}^{|\mathcal{X}|}$ takes an explanation target function $\gamma_{f,*}$ and an explicand $\boldsymbol{x}^e$ as inputs. The attribution method returns $\phi(\gamma_{f,*}, \boldsymbol{x}^e) \in \mathbb{R}^{|\mathcal{X}|}$ as feature importance scores. For $k = 1, ..., |\mathcal{X}|$, let $\boldsymbol{x}_k^e$ denote the $k$th feature of $\boldsymbol{x}^e$ and $\phi_k(\gamma_{f,*}, \boldsymbol{x}^e)$ the corresponding feature importance of $\boldsymbol{x}_k^e$. We demonstrate this definition of local feature attribution methods with the following example.

**Example 2.2.** *Given an explicand $\boldsymbol{x}^e$, the above classifier $f^{class}$, and the explanation target function in **Example 2.1**, consider Vanilla Gradients (Simonyan et al., 2013) as the local feature attribution method. The feature importance scores of $\boldsymbol{x}^e$ are computed as*

$$\phi(\gamma_{f^{class},\boldsymbol{x}^e}, \boldsymbol{x}^e) = \nabla_{\boldsymbol{x}} \gamma_{f^{class},\boldsymbol{x}^e}(\boldsymbol{x})\Big|_{\boldsymbol{x}=\boldsymbol{x}^e} = \nabla_{\boldsymbol{x}} f^{class}_{\arg\max_{j=1,...,C} f^{class}_j(\boldsymbol{x}^e)}(\boldsymbol{x})\Big|_{\boldsymbol{x}=\boldsymbol{x}^e}.$$

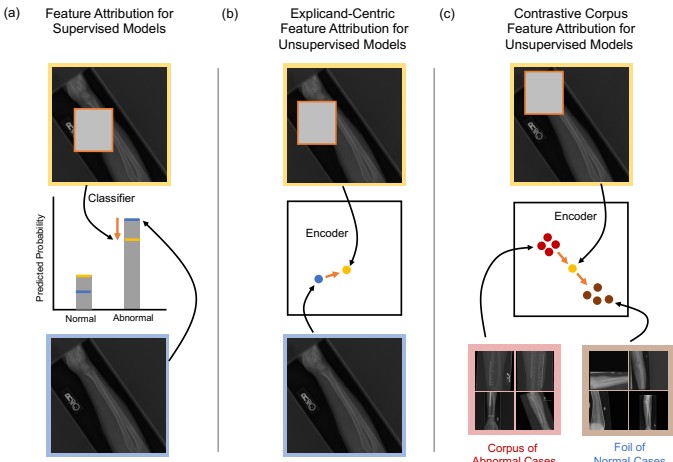

Figure 1: Illustration of feature attribution approaches in different settings. (a) Feature attribution for a supervised model (e.g., an X-ray image classifier for bone abnormality) identifies features that, when removed, decrease the predicted probability of a class of interest (e.g., abnormal). (b) Explicand-centric feature attribution (i.e., RELAX and label-free feature importance) for an unsupervised model identifies important features that, when removed, make the representation dissimilar to the explicand's original representation. (c) Contrastive corpus attribution for an unsupervised model identifies important features that, when removed, make the representation similar to a foil set (e.g., normal cases) and dissimilar to a corpus set (e.g., abnormal cases).

## 3  APPROACH

In this section, we motivate and propose a novel explanation target function for local feature attributions in the setting of explaining representations learned by an encoder model. We also describe how to combine the proposed explanation target function with existing attribution methods and how to interpret the obtained attribution scores. All proofs are in Appendix H.

### 3.1  DEFINING CONTRASTIVE CORPUS SIMILARITY AS AN EXPLANATION TARGET FUNCTION

We first define the notion of similarity between representations. Without loss of generality, suppose the representation space is $\mathbb{R}^d$ for some integer $d > 0$.

**Definition 3.1** (Representation similarity). *Let $f : \mathcal{X} \to \mathbb{R}^d$ be a representation encoder. For all $\boldsymbol{x}, \boldsymbol{x}' \in \mathcal{X}$, the representation similarity between $\boldsymbol{x}$ and $\boldsymbol{x}'$ based on $f$ is*

$$s_f(\boldsymbol{x}, \boldsymbol{x}') = \frac{f(\boldsymbol{x})^T f(\boldsymbol{x}')}{\|f(\boldsymbol{x})\|\|f(\boldsymbol{x}')\|}, \tag{1}$$

*where $\|\cdot\|$ denotes the Euclidean norm of a vector.*

We note that the above representation similarity is the cosine similarity of representations. Motivations for this choice are in Appendix B.

With **Definition 3.1**, the explanation target functions for RELAX and label-free feature importance are $\gamma_{f,\boldsymbol{x}^e}(\cdot) = s_f(\cdot, \boldsymbol{x}^e)$ and $\gamma_{f,\boldsymbol{x}^e}(\cdot) = s_f(\cdot, \boldsymbol{x}^e)\|f(\cdot)\|\|f(\boldsymbol{x}^e)\|$, respectively (Wickstrøm et al., 2021; Crabbé & van der Schaar, 2022). Intuitively, these explanation target functions identify explicand features that, when removed or perturbed, point the altered representation away from the explicand's original representation.

As mentioned in Section 1, we address the limitations of prior works by (i) relaxing the representation reference for feature attribution to any *corpus* set of samples; and (ii) enabling explicit specification of a foil to answer contrastive questions. For generality, we define the foil to be a distribution of inputs, which can also remove the extra burden of identifying particular foil samples. Nevertheless, we acknowledge that a fixed set of foil samples can indeed be selected for tailored use cases. Overall, our proposed improvements lead to *contrastive corpus similarity* as defined below.

**Definition 3.2** (Contrastive corpus similarity). *Let $\mathcal{C} \subset \mathcal{X}$ be a finite set of corpus samples and $\mathcal{D}_{foil}$ be a foil distribution over $\mathcal{X}$. The contrastive corpus similarity of any $\boldsymbol{x} \in \mathcal{X}$ to $\mathcal{C}$ in contrast to $\mathcal{D}_{foil}$ is*

$$\gamma_{f,\mathcal{C},\mathcal{D}_{foil}}(\boldsymbol{x}) = \frac{1}{|\mathcal{C}|}\sum_{\boldsymbol{x}^c \in \mathcal{C}} s_f(\boldsymbol{x}, \boldsymbol{x}^c) - \mathbb{E}_{\boldsymbol{x}^v \sim \mathcal{D}_{foil}}[s_f(\boldsymbol{x}, \boldsymbol{x}^v)]. \tag{2}$$

*Given a set of foil samples $\mathcal{F} = \{\boldsymbol{x}^{(1)}, ..., \boldsymbol{x}^{(m)}\}$, where $\boldsymbol{x}^{(1)}, ..., \boldsymbol{x}^{(m)} \overset{i.i.d.}{\sim} \mathcal{D}_{foil}$, the empirical estimator for the contrastive corpus similarity is*

$$\hat{\gamma}_{f,\mathcal{C},\mathcal{F}}(\boldsymbol{x}) = \frac{1}{|\mathcal{C}|}\sum_{\boldsymbol{x}^c \in \mathcal{C}} s_f(\boldsymbol{x}, \boldsymbol{x}^c) - \frac{1}{m}\sum_{i=1}^{m} s_f(\boldsymbol{x}, \boldsymbol{x}^{(i)}). \tag{3}$$

The unbiasedness of and a concentration bound for the empirical estimator are stated in Appendix C. Practically, these analyses allow us to choose an empirical foil set size with theoretical justification.

## 3.2 Generating and interpreting feature attributions

As in **Example 2.2**, given a feature attribution method $\phi : \mathbb{R}^{\mathcal{X}} \times \mathcal{X} \to \mathbb{R}^{|\mathcal{X}|}$ which takes an explanation target function $\gamma_{f,*}$ and an explicand $\boldsymbol{x}^e$ as inputs, COCOA computes attributions by explaining the empirical contrastive corpus similarity:

$$\phi(\hat{\gamma}_{f,\mathcal{C},\mathcal{F}}, \boldsymbol{x}^e) \tag{4}$$

In practice, this amounts to wrapping the representation encoder $f$ with the empirical estimator for contrastive corpus similarity and applying a feature attribution method. In particular, there are two classes of feature attribution methods which can be used to explain contrastive corpus similarity and other similarity-based explanation target functions. The first are removal-based methods (e.g., RISE, KernelSHAP) and the second are gradient-based methods (e.g., Vanilla Gradients, Integrated Gradients). Since the explanation target functions we consider in this paper can always be evaluated and are differentiable with respect to the inputs, we can directly apply these methods without modification. Finally, it is worth noting that methods which satisfy desirable properties (Sundararajan et al., 2017; Sundararajan & Najmi, 2020) in the supervised feature attribution setting also satisfy analogous properties for similarity-based explanation targets (Appendix D).

To interpret feature attribution results from COCOA, consider the *contrastive direction* toward the corpus representation mean and away from the foil representation mean. Features with high COCOA scores have a large impact on the explicand's representation direction pointing along this contrastive direction. This interpretation of COCOA is formalized in the following proposition.

**Proposition 3.3.** *The empirical estimator of contrastive corpus similarity is the dot product of the input's representation direction and the empirical contrastive direction. That is,*

$$\hat{\gamma}_{f,\mathcal{C},\mathcal{F}}(\boldsymbol{x}) = \left(\frac{f(\boldsymbol{x})}{\|f(\boldsymbol{x})\|}\right)^T \left(\frac{1}{|\mathcal{C}|}\sum_{\boldsymbol{x}^c \in \mathcal{C}}\frac{f(\boldsymbol{x}^c)}{\|f(\boldsymbol{x}^c)\|} - \frac{1}{m}\sum_{i=1}^{m}\frac{f(\boldsymbol{x}^{(i)})}{\|f(\boldsymbol{x}^{(i)})\|}\right). \tag{5}$$

**Proposition E.1** is an analogous statement for contrastive corpus similarity. An extension of **Proposition 3.3** and **Proposition E.1** to general kernel functions is in Appendix F. This extension implies that the directional interpretation of COCOA generally holds for kernel-based similarity functions.

## 4 EXPERIMENTS

In this section, we quantitatively evaluate whether important features identified by COCOA are indeed related to the corpus and foil, using multiple datasets, encoders, and feature attribution methods (Section 4.1). Furthermore, we demonstrate the utility of COCOA by its application to understanding image data augmentations (Section 4.2) and to mixed modality object localization (Section 4.3).

### 4.1 QUANTITATIVE EVALUATION

To evaluate COCOA across different representation learning models and datasets, we apply them to (i) SimCLR, a contrastive self-supervised model (Chen et al., 2020), trained on ImageNet (Russakovsky et al., 2015); (ii) SimSiam, a non-contrastive self-supervised model (Chen & He, 2021), trained on CIFAR-10 (Krizhevsky et al., 2009); and (iii) representations extracted from the penultimate layer of a ResNet18 (He et al., 2016), trained on the abnormal-vs.-normal musculoskeletal X-ray dataset MURA (Rajpurkar et al., 2017).

**Setup.** We consider four explanation targets: representation similiarity, contrastive similarity, corpus similarity, and contrastive corpus similarity; in our tables, these correspond to the methods: label-free, contrastive label-free, corpus, and COCOA, respectively. To conduct ablation studies for the components of COCOA, we introduce *contrastive similarity*, which is contrastive corpus similarity when the corpus similarity term is replaced with similarity to an explicand; and *corpus similarity*, which is contrastive corpus similarity without a foil similarity term. We use cosine similarity in all methods for the main text results (analogous results with dot product similarity are in Appendix M).

To evaluate the feature attributions for a given explicand image $x^e \in \mathbb{R}^{h,w,c}$ and explanation target, we compute feature attributions using either Integrated Gradients (Sundararajan et al., 2017), GradientSHAP (Smilkov et al., 2017; Erion et al., 2021), or RISE (Petsiuk et al., 2018). Then, we average the attributions across the channel dimension in order to use insertion and deletion metrics (Petsiuk et al., 2018). For deletion, we start with the original explicand and "remove" the $M$ most important pixels by masking[1] them with a mask $m \in \{0,1\}^{h,w,c}$ and a blurred version of the explicand $x^m \in \mathbb{R}^{h,w,c}$. We then evaluate the impact of each masked image with an evaluation measure $\eta(m \odot x^e + (1-m) \odot x^m)$. In the supervised setting, this evaluation measure is typically the predicted probability of a class of interest. If we plot the number of removed features on the x-axis and the evaluation measure on the y-axis, we would expect the curve to drop sharply initially, leading to a low area under the curve. For insertion, which "adds" important features, we would expect a high area under the curve. In the setting of explaining representations, we measure (i) the contrastive corpus similarity, which is calculated with the same corpus and foil sets used for explanations; and (ii) the corpus majority prediction, which is the predicted probability of the majority class based on each corpus sample's prediction, where the prediction is from a downstream linear classifier trained to predict classes based on representations[2].

We evaluate and average our insertion and deletion metrics for 250 explicands drawn for ImageNet and CIFAR-10 and 50 explicands for MURA, all from held-out sets. We consider both scenarios in which explicands and the corpus are in the same class and from different classes. For our experiments, 100 training samples are randomly selected to be the corpus set for each class in CIFAR-10 and MURA. For ImageNet, 100 corpus samples are drawn from each of the 10 classes from ImageNette (Howard, 2019) instead of all 1,000 classes for computational feasibility. Foil samples are randomly drawn from each training set. Because ResNets output non-negative representations from ReLU activations, we can apply Equation (C.3) and set the foil size to 1,500 based on a $\delta = 0.01$ and $\varepsilon = 0.05$ in **Proposition C.2**. We find that, even with a corpus size = 1, COCOA can have good performance, and results with corpus size $\geq 20$ are similar to each other (Appendix J). Varying the foil size does not seem to impact COCOA performance (Appendix K). Additional experiment details are in Appendix I.

**Results.** First, our evaluations show that COCOA consistently has the strongest performance in contrastive corpus similarity insertion and deletion metrics across all feature attribution methods, models, and datasets (Appendix L). These metrics imply that COCOA generates feature attributions that successfully describe whether pixels in the explicand make it more similar to the corpus and

---

[1]We always mask out all channels for a given pixel together.

[2]Except for MURA, where we use the final layer of the jointly trained network.

Table 1: Insertion and deletion metrics of corpus majority probability when explicands belong to the corpus class. Means (95% confidence intervals) across 5 experiment runs are reported. Higher insertion and lower deletion values indicate better performance, respectively. Each method is a combination of a feature attribution method and an explanation target function (e.g., COCOA under RISE corresponds to feature attributions computed by RISE for the contrastive corpus similarity). Performance results of random attributions (last row) are included as benchmarks.

| Attribution Method | Imagenet & SimCLR | | CIFAR-10 & SimSiam | | MURA & ResNet | |
|---|---|---|---|---|---|---|
| | Insertion ($\uparrow$) | Deletion ($\downarrow$) | Insertion ($\uparrow$) | Deletion ($\downarrow$) | Insertion ($\uparrow$) | Deletion ($\downarrow$) |
| Integrated Gradients | | | | | | |
| Label-Free | 0.362 (0.005) | 0.136 (0.004) | **0.403 (0.010)** | 0.249 (0.007) | 0.631 (0.040) | 0.513 (0.027) |
| Contrastive Label-Free | 0.377 (0.005) | 0.125 (0.003) | 0.401 (0.011) | 0.243 (0.007) | 0.690 (0.019) | 0.453 (0.025) |
| Corpus | 0.377 (0.005) | 0.147 (0.002) | 0.355 (0.014) | 0.249 (0.010) | 0.653 (0.014) | 0.500 (0.039) |
| COCOA | **0.422 (0.006)** | **0.119 (0.003)** | 0.386 (0.012) | **0.230 (0.011)** | **0.807 (0.013)** | **0.330 (0.030)** |
| Gradient SHAP | | | | | | |
| Label-Free | 0.409 (0.004) | 0.131 (0.001) | 0.500 (0.008) | 0.244 (0.013) | 0.691 (0.038) | 0.523 (0.033) |
| Contrastive Label-Free | 0.411 (0.003) | 0.127 (0.002) | 0.500 (0.009) | 0.238 (0.012) | 0.697 (0.037) | 0.510 (0.018) |
| Corpus | 0.421 (0.006) | 0.136 (0.001) | 0.478 (0.008) | 0.242 (0.009) | 0.729 (0.024) | 0.494 (0.037) |
| COCOA | **0.445 (0.003)** | **0.123 (0.002)** | **0.508 (0.007)** | **0.211 (0.008)** | **0.788 (0.030)** | **0.419 (0.013)** |
| RISE | | | | | | |
| Label-Free (RELAX) | 0.396 (0.005) | 0.160 (0.005) | 0.630 (0.005) | 0.283 (0.004) | 0.704 (0.022) | 0.600 (0.012) |
| Contrastive Label-Free | 0.424 (0.008) | 0.141 (0.005) | 0.632 (0.008) | 0.279 (0.004) | 0.730 (0.019) | 0.534 (0.015) |
| Corpus | 0.394 (0.010) | 0.166 (0.003) | 0.588 (0.005) | 0.314 (0.006) | 0.701 (0.019) | 0.617 (0.026) |
| COCOA | **0.456 (0.009)** | **0.126 (0.001)** | **0.663 (0.006)** | **0.256 (0.006)** | **0.840 (0.009)** | **0.415 (0.025)** |
| Random | 0.269 (0.003) | 0.268 (0.002) | 0.329 (0.013) | 0.329 (0.010) | 0.624 (0.018) | 0.629 (0.018) |

Table 2: Insertion and deletion metrics of corpus majority probability when explicands *do not* belong to the corpus class. Means (95% confidence intervals) across 5 experiment runs are reported. Higher insertion and lower deletion values indicate better performance, respectively. Each method is a combination of a feature attribution method and an explanation target function (e.g., COCOA under RISE corresponds to feature attributions computed by RISE for the contrastive corpus similarity). Performance results of random attributions (last row) are included as benchmarks.

| Attribution Method | Imagenet & SimCLR | | CIFAR-10 & SimSiam | | MURA & ResNet | |
|---|---|---|---|---|---|---|
| | Insertion ($\uparrow$) | Deletion ($\downarrow$) | Insertion ($\uparrow$) | Deletion ($\downarrow$) | Insertion ($\uparrow$) | Deletion ($\downarrow$) |
| Integrated Gradients | | | | | | |
| Label-Free | 3.53e-04 $\pm$ 1.06e-04 | 4.44e-04 $\pm$ 1.01e-04 | 0.061 (0.004) | 0.079 (0.004) | 0.394 (0.028) | 0.481 (0.031) |
| Contrastive Label-Free | 3.36e-04 $\pm$ 1.13e-04 | 4.44e-04 $\pm$ 1.00e-04 | 0.062 (0.004) | 0.082 (0.004) | 0.354 (0.022) | 0.518 (0.027) |
| Corpus | 1.09e-03 $\pm$ 2.85e-04 | 2.26e-04 $\pm$ 5.31e-05 | 0.094 (0.003) | 0.066 (0.003) | 0.609 (0.017) | 0.262 (0.029) |
| COCOA | **1.69e-03 $\pm$ 5.07e-04** | **1.55e-04 $\pm$ 2.21e-05** | **0.099 (0.004)** | **0.059 (0.005)** | **0.647 (0.017)** | **0.213 (0.030)** |
| Gradient SHAP | | | | | | |
| Label-Free | 2.05e-04 $\pm$ 5.96e-05 | 5.55e-04 $\pm$ 1.11e-04 | 0.054 (0.004) | 0.080 (0.004) | 0.362 (0.031) | 0.469 (0.021) |
| Contrastive Label-Free | 2.02e-04 $\pm$ 5.06e-05 | 5.10e-04 $\pm$ 9.65e-05 | 0.053 (0.002) | 0.080 (0.004) | 0.361 (0.022) | 0.477 (0.018) |
| Corpus | 8.03e-04 $\pm$ 1.64e-04 | 2.65e-04 $\pm$ 5.44e-05 | 0.106 (0.006) | 0.055 (0.003) | 0.549 (0.020) | 0.325 (0.019) |
| COCOA | **1.56e-03 $\pm$ 4.32e-04** | **1.67e-04 $\pm$ 5.66e-05** | **0.122 (0.008)** | **0.045 (0.003)** | **0.592 (0.025)** | **0.236 (0.012)** |
| RISE | | | | | | |
| Label-Free (RELAX) | 3.70e-04 $\pm$ 7.34e-05 | 5.37e-04 $\pm$ 1.23e-04 | 0.039 (0.003) | 0.080 (0.005) | 0.330 (0.016) | 0.433 (0.030) |
| Contrastive Label-Free | 3.32e-04 $\pm$ 1.14e-04 | 5.84e-04 $\pm$ 8.24e-05 | 0.040 (0.003) | 0.081 (0.005) | 0.307 (0.018) | 0.472 (0.027) |
| Corpus | 4.79e-04 $\pm$ 9.68e-05 | 3.87e-04 $\pm$ 8.41e-05 | 0.086 (0.005) | 0.043 (0.003) | 0.497 (0.031) | 0.269 (0.020) |
| COCOA | **9.51e-04 $\pm$ 2.58e-04** | **3.60e-04 $\pm$ 1.40e-04** | **0.107 (0.007)** | **0.031 (0.002)** | **0.590 (0.027)** | **0.181 (0.014)** |
| Random | 4.87e-04 $\pm$ 9.50e-05 | 5.03e-04 $\pm$ 9.73e-05 | 0.070 (0.003) | 0.070 (0.004) | 0.406 (0.013) | 0.407 (0.016) |

dissimilar to the foil based on cosine similarity. Although this is an important sanity check, it is perhaps unsurprising that COCOA performs strongly in this evaluation metric, because COCOA's explanation target is exactly the same as this evaluation metric.

Next, in Tables 1 and 2, COCOA has the strongest performance in corpus majority probability insertion and deletion metrics across nearly all feature attribution methods, models, and datasets. This is arguably a more interesting evaluation compared to contrastive corpus similarity, given that COCOA is not explicitly designed to perform this task. Evaluation using the corpus majority predicted probability is semantically meaningful, because it tests whether features with high attributions move an

explicand representation closer to the corpus as opposed to other classes across a decision boundary. Furthermore, evaluation with a linear classifier has direct implications for downstream classification tasks and follows the widely used linear evaluation protocol for representation learning (Bachman et al., 2019; Kolesnikov et al., 2019; Chen et al., 2020). In Table 1, we evaluate each method in the setting where explicands are from the same class as the corpus. This enables us to compare to label-free attribution methods, which can still perform strongly on these metrics, because large changes in the explicand representation likely correlate with changes in the predicted probability of the explicand's class. Here, we can see that COCOA consistently outperforms label-free attribution methods and also benefits from the combination of a corpus and foil set. Importantly, in Table 2, corpus-based attribution methods perform strongly especially when the corpus is from a different class compared to the explicand. This is significant, because it implies that we can use COCOA to generate explanations that answer questions such as, "What pixels in my dog image make my representation similar to cat images?", which was previously impossible using label-free approaches.

## 4.2 UNDERSTANDING DATA AUGMENTATIONS IN SIMCLR

In this section, we aim to visualize augmentation-invariant features that preserve class labels in SimCLR. Because class labels are associated with original images, we use COCOA to find features that make the representation of an augmented image similar to that of the original image.

SimCLR is a self-supervised contrastive learning method that pushes images from the same class close together and images from different classes far apart. This is achieved by its contrastive objective, which maximizes the similarity between representations from two augmentations of the same image and minimizes the similarity between representations from different images. Empirically, linear classifiers trained on representations learned by SimCLR perform comparably to strong supervised baselines (Chen et al., 2020). Theoretical analyses of SimCLR performance rely on the idea that data augmentations preserve the class label of an image (Arora et al., 2019; HaoChen et al., 2021). Here, we show which input features are associated with label preservation, which has not been previously shown.

**Setup.** The same SimCLR model and its downstream classifier for ImageNet from Section 4.1 are used here. For each image, we investigate augmentations used to train SimCLR including flipping, cropping, flipping with cropping, grayscaling, and color jittering. We also consider cutout and $90°$ rotation, which are not included in SimCLR training. Class prediction based on the linear classifier is computed for the original version and all augmentations of each image to check class preservation. With each original and augmented image as the explicand, the original image as the corpus, and 1,500 random images from the ImageNet training set as the foil, COCOA is paired with RISE to visually identify class-preserving features. The RISE parameters are the same as in Section 4.1.

**Results.** We first note that the original images and the augmented images after flipping, cropping, flipping with cropping, grayscaling, and color jittering yield correct class predictions (Figure 2). However, cutout and rotated images, which are not part of SimCLR training, are less robust and can result in misclassifications. Because class labels are generated based on the original images, important features identified by COCOA with each original image as the corpus are class-preserving for correctly classified augmentations. Qualitatively, the class-preserving region of the English springer seems to be its face, as the face is highlighted in all augmentations with the correct classification and the original image (Figure 2a). Contrarily, in the rotated image, which is misclassified as a llama, COCOA highlights the body of the English springer instead of its face. Similarly, the piping in the French horn is the class-preserving region (Figure 2b). When the piping is cut out, the image is misclassified, and COCOA highlights only the face of the performer. Finally, the co-occurrence of the parachute and parachuter is the important feature for an augmented image to have the correct classification of parachute (Figure 2c). Additional results are in Appendix O.

## 4.3 MIXED MODALITY OBJECT LOCALIZATION

Here, we showcase the utility of COCOA by performing zero-shot object localization based on mixed modality representations as in Gadre et al. (2022). In this experiment, we work with CLIP which trains a text and image encoder on paired data: images with associated captions (Radford et al., 2021). To apply COCOA, a necessary prerequisite is a set of corpus examples. These examples are not necessarily hard to gather, since small sets of corpus examples can be sufficient (Appendix J).

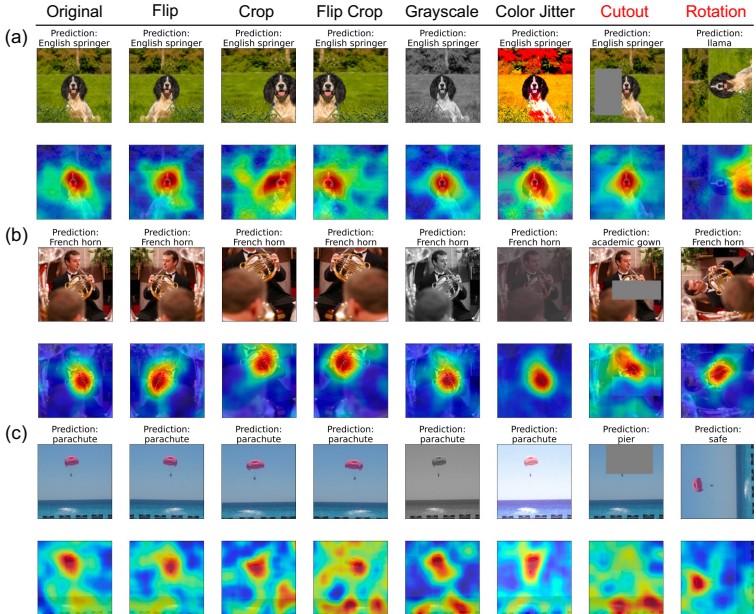

Figure 2: Original version and augmentations of images with their class predictions (top row), along with the corresponding COCOA attributions (red for higher values and blue for lower values) with each original image as the corpus and random images as the foil (bottom row); for (a) an English springer, (b) a French horn, and (c) a parachute. Cutout and rotation are not included in SimCLR training.

However, if our image representations are generated via the CLIP image encoder, we can circumvent the need to gather a set of corpus images by instead explaining similarity of our explicand image to a corpus caption of interest. This enables us to easily ask semantically meaningful questions such as, "What in this image makes its representation similar to a woman?".

**Setup.** In this experiment, we have two CLIP representation encoders $f^{text} : \mathcal{X}^{text} \to \mathbb{R}^d$ and $f^{image} : \mathcal{X}^{image} \to \mathbb{R}^d$. Both encoders map their respective input domains into a common representation space such that the representations of a caption and image pair have high cosine similarity when the caption is descriptive of the paired image. We use the original implementation of CLIP[3] and model ViT-B/32. The feature attributions are computed using RISE with the same parameters as in Section 4.1, except with 20,000 masks.

Here, we aim to explain an explicand image $\boldsymbol{x}^e \in \mathcal{X}^{image}$. For label-free importance, the explanation target is the representation similarity based on the image encoder: $\gamma_{f^{image}, \boldsymbol{x}^e}(\cdot)$. Then, we use an adapted version of the COCOA explanation target, which uses a text corpus set $\mathcal{C}^{text} \subset \mathcal{X}^{text}$ and a text foil set $\mathcal{F}^{text} \subset \mathcal{X}^{text}$ and maps from images to contrastive corpus similarity $\hat{\gamma}_{f^{image}, f^{text}, \mathcal{C}^{text}, \mathcal{F}^{text}}(\boldsymbol{x}) : \mathcal{X}^{image} \to \mathbb{R}$, defined as follows:

$$\frac{1}{|\mathcal{C}^{text}|} \sum_{\boldsymbol{x}^c \in \mathcal{C}^{text}} \frac{f^{image}(\boldsymbol{x})^T f^{text}(\boldsymbol{x}^c)}{\|f^{image}(\boldsymbol{x})\| \|f^{text}(\boldsymbol{x}^c)\|} - \frac{1}{|\mathcal{F}^{text}|} \sum_{\boldsymbol{x}^f \in \mathcal{F}^{text}} \frac{f^{image}(\boldsymbol{x})^T f^{text}(\boldsymbol{x}^f)}{\|f^{image}(\boldsymbol{x})\| \|f^{text}(\boldsymbol{x}^f)\|}. \quad (6)$$

In particular, the corpus set consists of a single caption $\mathcal{C} = \{\boldsymbol{x}^c\}$ of the following form, $\boldsymbol{x}^c =$ "This is a photo of a $P$", where $P$ is manually chosen. Then, the foil set is either (i) a single caption $\mathcal{F} = \{\boldsymbol{x}^f\}$ of the form, $\boldsymbol{x}^f =$ "This is a photo of a $Q$", where $Q$ is manually chosen or (ii) many captions $\mathcal{F} = \{\boldsymbol{x}_i^f\}$ of the forms, $\boldsymbol{x}_i^f =$ "This is a photo of a $Q_i$" where $Q_i \in \mathcal{C}^{cifar100} \setminus P$ is each of the 100 classes in CIFAR-100 excluding the corpus class $P$. Each explicand image is a public domain image from https://www.pexels.com/ that the CLIP model is unlikely to have been trained on[4].

---

[3]https://github.com/openai/CLIP

[4]With the exception of the astronaut image, which is available in the scikit-image package.

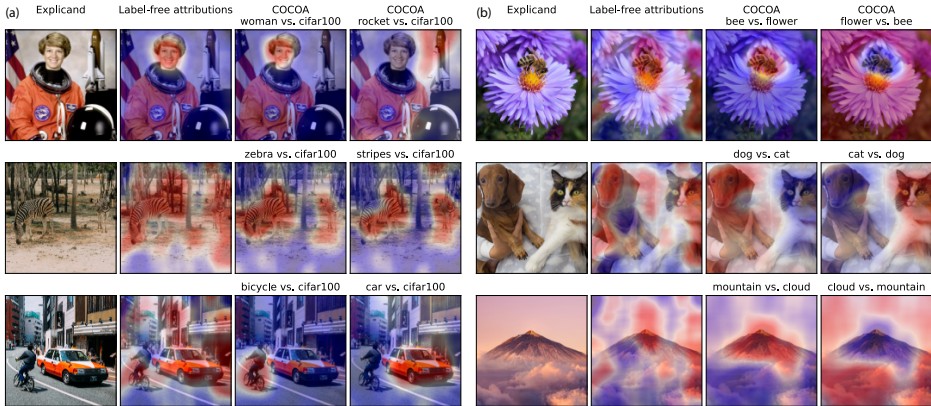

Figure 3: Visualization of feature attributions. (a) Importance of a corpus text compared to many foil texts (based on CIFAR-100 classes). (b) Importance of a corpus text compared to a foil text. (a)-(b) The leftmost column is the original explicand, the second column is label-free attributions, and the third and fourth columns are COCOA attributions using $P$ vs. $Q$ as the notation for the corpus and foil text sets (details in Section 4.3). Each feature attribution is overlaid on the original image such that blue is the minimum value and red is the maximum value in given attribution.

**Results.** We visualize the label-free and COCOA attributions for several explicands in Figure 3. First, we find that label-free attributions highlight a variety of regions in the explicand image. Although these pixels contribute to similarity to the explicand's representation, with the exception of the astronaut image, the important pixels are largely not localized and therefore hard to interpret. For instance, in the mountain image we cannot say whether the mountain or cloud is the most important to the explicand's representation, because the most positively and negatively important pixels are distributed across both objects.

Instead, we find that using COCOA, we can ask specific questions. For instance, we may ask, "Compared to many other classes, what pixels in this image look like $P$?" (Figure 3a). We evaluate a number of these questions in the first three images. In the astronaut image, we can see the woman's face and the rocket are localized. Likewise in the image with both a bicycle and a car, we can localize either the bicycle or the car. In the zebra image, we are able to localize the zebras by using corpus texts that aim to identify a "zebra" or "stripes". Finally, in the last three images we find that we can formulate specific contrastive questions that ask, "What pixels in this image look like $P$ rather than $Q$?" (Figure 3b). Doing so we successfully identify a bee rather than a flower, a dog rather than a cat, and a mountain rather than clouds (and vice versa respectively).

## 5 DISCUSSION

In this work, we introduce COCOA, a method to explain unsupervised models based on similarity in a representation space. We theoretically analyze our method and demonstrate its compatibility with many feature attribution methods. We quantitatively validate COCOA in several datasets and on contrastive self-supervised, non-contrastive self-supervised, and supervised models. Then, we use COCOA to better understand data augmentations in a self-supervised setting and to perform zero-shot object localization.

Although COCOA is already a powerful technique to understand representations, there are many interesting future research directions. First, we can adapt automatic concept discovery methods (Ghorbani et al., 2019; Yeh et al., 2020) to automatically identify homogeneous corpus sets. Second, we can use other explanation methods for our novel contrastive corpus similarity explanation target. This includes counterfactual explanation methods (Verma et al., 2020), which only require gradients to optimize a counterfactual objective. Finally, a fundamental limitation of COCOA is that it cannot explain corpus sets if the model has not learned anything about the given corpus. Therefore, a valuable future direction is to investigate complementary methods to detect whether the model's representation contains information about a corpus prior to application of COCOA.

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

## A    ADDITIONAL RELATED WORK

**Representation learning.** COCOA can be used to explain any representation learning model. Representation learning aims to automatically discover latent dimensions that are valuable to downstream tasks (e.g., classification) (Bengio et al., 2013). It has been applied to great effect within natural language processing (Deng et al., 2013; Van Den Oord et al., 2017) and computer vision (Ciregan et al., 2012; Gidaris et al., 2018) and encompasses both supervised and unsupervised models ($f : \mathcal{X} \rightarrow \mathcal{Z}$). For supervised models the output space corresponds to known labels ($\mathcal{Z} = \mathcal{Y}$), whereas for unsupervised models the outputs are latent representations ($\mathcal{Z} = \mathcal{H}$). Within unsupervised representation learning, self-supervised learning leverages unlabelled data to formulate learning tasks (Grill et al., 2020; Misra & Maaten, 2020). One particularly successful instantiation of self-supervised learning is contrastive learning, which optimizes the similarity between different views of the same sample and dissimilarity to random samples (Chen et al., 2020; Radford et al., 2021).

**Representation-space explanations.** COCOA explains a representation (latent) space. An early post-hoc explanation method based on representation spaces is TCAV (Kim et al., 2018) which defines concepts based on the representation space and produces a global explanation of whether a given class depends on a concept of interest. A number of approaches similarly define concepts in the representation space and build upon TCAV with automatic concept discovery (Ghorbani et al., 2019; Yeh et al., 2020). Finally, Basaj et al. (2021) build upon TCAV to define visual probes to validate whether a representation encodes a property of interest. These approaches detect whether a given concept is important to the model, whereas our approach aims to understand which features make a representation similar to a given concept captured by a corpus.

**Supervised explanations.** COCOA builds upon existing supervised explanation methods to explain an unsupervised setting. The majority of explanation methods operate on supervised models and aim to explain a scalar output, often the model's predicted probability for a specific class. These methods vary in the unit of explanation (i.e., the way in which they describe what was important) and the explanation target (i.e., the model behavior they aim to explain). The unit of explanation for example attributions and counterfactual attributions are samples which are either important to or modify the explanation target (Koh & Liang, 2017; Wachter et al., 2017). In contrast, the unit of explanation of feature attributions or concept attributions are importances which quantify how features or concepts contribute to the explanation target (Lundberg & Lee, 2017; Kim et al., 2018).

In this paper, we focus on feature attribution methods, which are the most widely-used explanation technique, but note that it is possible to apply many other explanation techniques to understand our new explanation target. It has been shown that different feature attribution methods can generate disagreeing results (Krishna et al., 2022), so we empirically evaluate contrastive corpus similarity and other explanation target functions with a number of varied attribution methods. In particular, we compute local feature attributions using three popular methods. The first is Integrated Gradients (Sundararajan et al., 2017), a gradient-based feature attribution method, which accumulates gradients between a baseline and an explicand. The second is GradientSHAP[5], which is a SHAP-related method that combines SmoothGrad (Smilkov et al., 2017) and Expected Gradients (Erion et al., 2021), a generalization of Integrated Gradients. Third, we use RISE (Petsiuk et al., 2018), a removal-based explanation method (Covert et al., 2021), which computes feature attributions by randomly masking portions of an explicand.

**Unsupervised explanations.** COCOA explicitly aims to provide explanations for unsupervised models. There is largely a deficit of explanation methods suitable to this task and we are aware of only two recent methods that address this problem: RELAX (Wickstrøm et al., 2021) and label-free explainability (Crabbé & van der Schaar, 2022). Both methods define an explanation target function based on similarity to the explicand, where RELAX uses the cosine kernel ($\gamma_{f,\boldsymbol{x}^e}(\boldsymbol{x}) = (f(\boldsymbol{x})^T f(\boldsymbol{x}^e))/(\|f(\boldsymbol{x})\|\|f(\boldsymbol{x}^e)\|)$) and label-free explainability uses the dot product ($\gamma_{f,\boldsymbol{x}^e}(\boldsymbol{x}) = f(\boldsymbol{x})^T f(\boldsymbol{x}^e)$). Then, both approaches utilize pre-existing local feature attribution methods to explain their respective explanation targets, where RELAX specfically utilizes RISE, and label-free explainability is an agnostic wrapper function. A final distinction between the two papers is that label-free explainability additionally provides example attributions which aim to identify important training examples. While COCOA, RELAX, and label-free explainability are motivated to

---

[5]https://captum.ai/api/gradient_shap.html

explain unsupervised models, they all can be applied to explain representations in general, whether obtained through unsupervised or supervised learning.

**Contrastive explanations.** COCOA makes unsupervised explanations more interpretable by making them contrastive. Contrastive explanations aim to understand an event $P$ (the fact) by contrasting it to another event $Q$ (the foil) (Jacovi et al., 2021). Social science studies recognize that contrastive explanations align with human intuition and reduce an explanation's complexity by ignoring factors common to both the fact and the foil (Kahneman & Miller, 1986; Lipton, 1990; Miller, 2019). One strategy to obtain contrastive explanations is to simply define an explanation target function which describes a contrastive model behavior (e.g., the difference between the model's prediction for the fact and the foil) (Jacovi et al., 2021).

## B    MOTIVATIONS FOR COSINE SIMILARITY AS THE REPRESENTATION SIMILARITY

The representation similarity defined in **Definition 3.1** is the cosine similarity of representations. There are multiple motivations for this choice. First, this representation similarity is used in RELAX as the explanation target function (Wickstrøm et al., 2021). Similarly, label-free feature importance uses the dot product without normalization (Crabbé & van der Schaar, 2022). Hence, the choice of cosine similarity facilitates comparisons with prior works. Second, many self-supervised models are trained with the cosine similarity or the related cross-correlation in their loss functions (Chen et al., 2020; Grill et al., 2020; Chen & He, 2021; Radford et al., 2021; Zbontar et al., 2021), so cosine similarity is a natural choice for measuring similarity between representations learned by these models. Finally, other similarity measures, such as the Gaussian kernel function, may contain parameters that impact explanation outcomes, but to our knowledge there are no principled ways of tuning these parameters for the purpose of generating explanations.

## C    SOME PROPERTIES OF THE EMPIRICAL ESTIMATOR OF CONTRASTIVE CORPUS SIMILARITY

**Proposition C.1.** *The estimator $\hat{\gamma}_{f,\mathcal{C},\mathcal{F}}(\boldsymbol{x})$ is an unbiased estimator of $\gamma_{f,\mathcal{C},\mathcal{D}_{foil}}(\boldsymbol{x})$.*

**Proposition C.2.** *For any $\delta \in (0,1)$ and $\varepsilon > 0$, if the sample size $m$ in $\hat{\gamma}_{f,\mathcal{C},\mathcal{F}}(\boldsymbol{x})$ satisfies*

$$m \geq \frac{2\log(2/\delta)}{\varepsilon^2}, \tag{C.1}$$

*then*

$$P(|\hat{\gamma}_{f,\mathcal{C},\mathcal{F}}(\boldsymbol{x}) - \gamma_{f,\mathcal{C},\mathcal{D}_{foil}}(\boldsymbol{x})| \geq \varepsilon) \leq \delta. \tag{C.2}$$

*If it is further assumed that $f$ maps from $\mathcal{X}$ to the set of non-negative vectors $\mathbb{R}^d_+$ (e.g., a neural network with a ReLU output layer), then the same concentration bound is achieved with*

$$m \geq \frac{\log(2/\delta)}{2\varepsilon^2}. \tag{C.3}$$

**Proposition C.2** allows us to choose an empirical foil sample size with theoretical justification.

## D    FEATURE ATTRIBUTION PROPERTIES

There are a multitude of feature attribution methods. In order to compare these methods, researchers have debated their respective merits in terms of the properties they satisfy (Sundararajan & Najmi, 2020). Most feature attribution methods satisfy certain properties with respect to the model output (explanation target) they explain. Methods which satisfy a property when the explanation target is the predicted probability of a particular class will also satisfy a related property for other explanation targets such as representation similarity or contrastive corpus similarity.

As an example, we may consider the property of completeness, which is satisfied by Integrated Gradients, KernelSHAP, DeepLIFT, and a number of other explanation methods. Methods which

satisfy completeness generate feature attributions which sum up to the explanation target function applied to the explicand minus a baseline value $b_0$:

$$\sum_{i=1}^{d} \phi_i(\gamma_{f,*}, \boldsymbol{x}^e) = \gamma_{f,*}(\boldsymbol{x}^e) - b_0. \tag{D.1}$$

For attribution methods that use a single baseline (e.g., Integrated Gradients, DeepLIFT), the baseline value is equal to the explanation target function applied to a baseline of the user's choosing (i.e., $b_0 = \gamma_{f,*}(\boldsymbol{x}^b)$). Therefore, supervised feature attributions sum up to the difference between the predicted probability of a class of interest for the explicand $\boldsymbol{x}^e$ and baseline $\boldsymbol{x}^b$:

$$\sum_{i=1}^{d} \phi_i(\gamma_{f^{class}}, \boldsymbol{x}^e) = \gamma_{f^{class}}(\boldsymbol{x}^e) - \gamma_{f^{class}}(\boldsymbol{x}^b). \tag{D.2}$$

Analogously, in the unsupervised explanation setting, COCOA feature attributions will sum up to the difference in the contrastive corpus similarity between the explicand and the baseline:

$$\sum_{i=1}^{d} \phi_i(\hat{\gamma}_{f,\mathcal{C},\mathcal{F}}, \boldsymbol{x}^e) = \hat{\gamma}_{f,\mathcal{C},\mathcal{F}}(\boldsymbol{x}^e) - \hat{\gamma}_{f,\mathcal{C},\mathcal{F}}(\boldsymbol{x}^b). \tag{D.3}$$

Note that analogous completeness properties may be shown for representation similarity, contrastive similarity, and corpus similarity.

## E  PROPOSITION FOR INTERPRETING COCOA WITH RESPECT TO THE CONTRASTIVE CORPUS SIMILARITY

**Proposition E.1.** *Contrastive corpus similarity is the dot product of the input's representation direction and the contrastive direction. That is,*

$$\gamma_{f,\mathcal{C},\mathcal{D}_{foil}}(\boldsymbol{x}) = \left( \frac{f(\boldsymbol{x})}{\|f(\boldsymbol{x})\|} \right)^T \left( \frac{1}{|\mathcal{C}|} \sum_{\boldsymbol{x}^c \in \mathcal{C}} \frac{f(\boldsymbol{x}^c)}{\|f(\boldsymbol{x}^c)\|} - \mathbb{E}_{\boldsymbol{x}^v \sim \mathcal{D}_{foil}} \left[ \frac{f(\boldsymbol{x}^v)}{\|f(\boldsymbol{x}^v)\|} \right] \right). \tag{E.1}$$

## F  EXTENSION OF **PROPOSITION 3.3** AND **PROPOSITION E.1** TO GENERAL KERNEL FUNCTIONS

Let another kernel function $s : \mathbb{R}^d \times \mathbb{R}^d \to \mathbb{R}$, such that $s_f(\cdot, \cdot) = s(f(\cdot), f(\cdot))$, be the representation similarity measure instead of the cosine similarity. Then, in another latent feature space, the contrastive direction points toward the corpus's mean representation and away from the foil's mean representation. Formally, we have the following proposition.

**Proposition F.1.** *Suppose the function $s(\cdot, \cdot)$ is a continuous positive semi-definite kernel on a compact set $\mathcal{Z} \subset \mathbb{R}^d$, and $f$ maps from $\mathcal{X}$ to $\mathcal{Z}$. Let $L_2(\mathcal{Z})$ denote the set of functions over $\mathcal{Z}$ that are square integrable. Additionally, suppose that the integral operator $T_s$, defined as $(T_s g)(\cdot) = \int_{\mathcal{Z}} s(\cdot, \tilde{z}) g(\tilde{z}) d\tilde{z}$ for all $g \in L_2(\mathcal{Z})$, is positive semi-definite:*

$$\int_{\mathcal{Z}} \int_{\mathcal{Z}} s(\boldsymbol{z}, \tilde{\boldsymbol{z}}) g(\tilde{\boldsymbol{z}}) d\boldsymbol{z} d\tilde{\boldsymbol{z}} \geq 0. \tag{F.1}$$

*Let*

$$\gamma_{f,\mathcal{C},\mathcal{D}_{foil}}(\boldsymbol{x}) = \frac{1}{|\mathcal{C}|} \sum_{\boldsymbol{x}^c \in \mathcal{C}} s(f(\boldsymbol{x}), f(\boldsymbol{x}^c)) - \mathbb{E}_{\boldsymbol{x}^v \sim \mathcal{D}_{foil}}[s(f(\boldsymbol{x}), f(\boldsymbol{x}^v))] \tag{F.2}$$

*and*

$$\hat{\gamma}_{f,\mathcal{C},\mathcal{F}}(\boldsymbol{x}) = \frac{1}{|\mathcal{C}|} \sum_{\boldsymbol{x}^c \in \mathcal{C}} s(f(\boldsymbol{x}), f(\boldsymbol{x}^c)) - \frac{1}{m} \sum_{i}^{m} s(f(\boldsymbol{x}), f(\boldsymbol{x}^{(i)})). \tag{F.3}$$

*Then*

$$\gamma_{f,\mathcal{C},\mathcal{D}_{foil}}(\boldsymbol{x}) = \psi(f(\boldsymbol{x}))^T \left( \frac{1}{|\mathcal{C}|} \sum_{\boldsymbol{x}^c \in \mathcal{C}} \psi(f(\boldsymbol{x}^c)) - \mathbb{E}_{\boldsymbol{x}^v \sim \mathcal{D}_{foil}}[\psi(f(\boldsymbol{x}^v))] \right) \tag{F.4}$$

*and*

$$\hat{\gamma}_{f,\mathcal{C},\mathcal{F}}(\boldsymbol{x}) = \psi(f(\boldsymbol{x}))^T \left( \frac{1}{|\mathcal{C}|} \sum_{\boldsymbol{x}^c \in \mathcal{C}} \psi(f(\boldsymbol{x}^c)) - \frac{1}{m} \sum_{i=1}^m \psi(f(\boldsymbol{x}^{(i)})) \right), \tag{F.5}$$

*where $\psi : \mathcal{Z} \to \mathcal{D}$ is a feature map from the representation space to the feature space $\mathcal{D}$ induced by the kernel function.*

## G  CONTRASTIVE CORPUS SIMILARITY WHEN THE CORPUS CONSISTS OF MULTIPLE SUB-CORPORA

Here, we consider the scenario where a corpus consists of multiple groups (i.e., *sub-corpora*) with different characteristics. In this setting, the contrastive corpus similarity and its empirical estimator are a weighted average of the contrastive sub-corpus similarities and a weighted average of the empirical estimators of contrastive sub-corpus similarities, respectively. Importantly, the weight corresponding to each sub-corpus is proportional to its size. This insight is formalized in the following proposition.

**Proposition G.1.** *Suppose a corpus $\mathcal{C}$ consists of disjoint sub-corpora $\mathcal{C}_k$ for $k = 1, ..., K$, for some integer $K > 0$. That is, $\mathcal{C} = \bigcup_{k=1}^K \mathcal{C}_k$, and $\mathcal{C}_i \cap \mathcal{C}_j = \emptyset$ for all $i \neq j$. Then the contrastive corpus similarity with respect to the corpus $\mathcal{C}$ is a weighted average of contrastive corpus similarities with respect to the sub-corpora:*

$$\gamma_{f,\mathcal{C},\mathcal{D}_{foil}}(\boldsymbol{x}) = \sum_{k=1}^K \frac{|\mathcal{C}_k|}{|\mathcal{C}|} \gamma_{f,\mathcal{C}_k,\mathcal{D}_{foil}}(\boldsymbol{x}). \tag{G.1}$$

*Furthermore, the empirical estimator of contrastive corpus similarity with respect to the corpus $\mathcal{C}$ is a weighted average of empirical estimators of contrastive corpus similarity with respect to the sub-corpora:*

$$\hat{\gamma}_{f,\mathcal{C},\mathcal{F}}(\boldsymbol{x}) = \sum_{k=1}^K \frac{|\mathcal{C}_k|}{|\mathcal{C}|} \hat{\gamma}_{f,\mathcal{C}_k,\mathcal{F}}(\boldsymbol{x}). \tag{G.2}$$

## H  PROOFS

### H.1  PROOF OF **PROPOSITION C.1**

*Proof.* We have

$$\mathbb{E}_{\boldsymbol{x}^{(1)},...,\boldsymbol{x}^{(m)} \overset{\text{i.i.d.}}{\sim} \mathcal{D}_{foil}}[\hat{\gamma}_{f,\mathcal{C},\mathcal{F}}(\boldsymbol{x})] = \frac{1}{|\mathcal{C}|} \sum_{\boldsymbol{x}^c \in \mathcal{C}} s_f(\boldsymbol{x}, \boldsymbol{x}^c) - \frac{1}{m} \sum_{i=1}^m \mathbb{E}_{\boldsymbol{x}^{(1)},...,\boldsymbol{x}^{(m)} \overset{\text{i.i.d.}}{\sim} \mathcal{D}_{foil}}[s_f(\boldsymbol{x}, \boldsymbol{x}^{(i)})] \tag{H.1}$$

$$= \frac{1}{|\mathcal{C}|} \sum_{\boldsymbol{x}^c \in \mathcal{C}} s_f(\boldsymbol{x}, \boldsymbol{x}^c) - \mathbb{E}_{x_v \sim \mathcal{D}_{foil}}[s_f(\boldsymbol{x}, \boldsymbol{x}^v)] = \gamma_{f,\mathcal{C},\mathcal{D}_{foil}}(\boldsymbol{x}), \tag{H.2}$$

where (H.1) follows from the linearity of expectation, and (H.2) from the fact that $\boldsymbol{x}^{(1)}, ..., \boldsymbol{x}^{(m)}$ are identically distributed from $\mathcal{D}_{foil}$. □

### H.2  PROOF OF **PROPOSITION C.2**

We first state Hoeffding's inequality.

**Lemma H.1.** *Let $X_1, ..., X_n$ be independent random variables such that $a_i \leq X_i \leq b_i$ almost surely. Consider $S_n = X_1 + \cdots + X_n$, then*

$$P(|S_n - \mathbb{E}[S_n]| \geq \varepsilon) \leq 2 \exp\left( -\frac{2\varepsilon^2}{\sum_{i=1}^n (b_i - a_i)^2} \right). \tag{H.3}$$

*Proof.* See Hoeffding (1963). □

We now proceed to prove **Proposition C.2**.

*Proof.* First, we have

$$P(|\hat{\gamma}_{f,\mathcal{C},\mathcal{F}}(\boldsymbol{x}) - \gamma_{f,\mathcal{C},\mathcal{D}_{foil}}(\boldsymbol{x})| \geq \varepsilon) = P\left( \left| \sum_{i=1}^{m} \frac{s_f(\boldsymbol{x}, \boldsymbol{x}^{(i)})}{m} - \mathbb{E}_{\boldsymbol{x}^v \sim \mathcal{D}_{foil}}[s_f(\boldsymbol{x}, \boldsymbol{x}^v)] \right| \geq \epsilon \right) \quad \text{(H.4)}$$

and note that

$$\mathbb{E}_{\boldsymbol{x}^v \sim \mathcal{D}_{foil}}[s_f(\boldsymbol{x}, \boldsymbol{x}^v)] = \mathbb{E}_{\boldsymbol{x}^{(1)}, \dots, \boldsymbol{x}^{(m)} \overset{\text{i.i.d.}}{\sim} \mathcal{D}_{foil}} \left[ \sum_{i=1}^{m} \frac{s_f(\boldsymbol{x}, \boldsymbol{x}^{(i)})}{m} \right]. \quad \text{(H.5)}$$

Because the cosine similarity is bounded: $-1 \leq s_f(\cdot, \cdot) \leq 1$, it follows that $-1/m \leq s_f(\cdot, \cdot)/m \leq 1/m$. Applying the Hoeffding's inequality, we obtain

$$P(|\hat{\gamma}_{f,\mathcal{C},\mathcal{F}}(\boldsymbol{x}) - \gamma_{f,\mathcal{C},\mathcal{D}_{foil}}(\boldsymbol{x})| \geq \varepsilon) \leq 2 \exp\left( -\frac{2\varepsilon^2}{\sum_{i=1}^{m}(2/m)^2} \right) \quad \text{(H.6)}$$

$$= 2 \exp\left( -\frac{\varepsilon^2}{2} \cdot m \right) \quad \text{(H.7)}$$

$$\leq 2 \exp\left( -\frac{\varepsilon^2}{2} \cdot \frac{2\log(2/\delta)}{\varepsilon^2} \right) = \delta. \quad \text{(H.8)}$$

If it is further assumed that $f$ maps from $\mathcal{X}$ to $\mathbb{R}_+^d$, then $0 \leq s_f(\cdot, \cdot) \leq 1$ and hence $0 \leq s_f(\cdot, \cdot)/m \leq 1/m$. Applying the Hoeffding's inequality with $m \geq \frac{\log(2/\delta)}{2\varepsilon^2}$ yields

$$P(|\hat{\gamma}_{f,\mathcal{C},\mathcal{F}}(\boldsymbol{x}) - \gamma_{f,\mathcal{C},\mathcal{D}_{foil}}(\boldsymbol{x})| \geq \varepsilon) \leq 2 \exp\left( -\frac{2\varepsilon^2}{\sum_{i=1}^{m}(1/m)^2} \right) \quad \text{(H.9)}$$

$$= 2 \exp(-2\varepsilon^2 \cdot m) \quad \text{(H.10)}$$

$$\leq 2 \exp\left( -2\varepsilon^2 \cdot \frac{\log(2/\delta)}{2\varepsilon^2} \right) = \delta. \quad \text{(H.11)}$$

□

## H.3 Proof of **Proposition 3.3** and **Proposition E.1**

*Proof.* By the definition of $\hat{\gamma}_{f,\mathcal{C},\mathcal{F}}(\boldsymbol{x})$ and linear algebra, we have

$$\hat{\gamma}_{f,\mathcal{C},\mathcal{F}}(\boldsymbol{x}) = \frac{1}{|\mathcal{C}|} \sum_{\boldsymbol{x}^c \in \mathcal{C}} \left( \frac{f(\boldsymbol{x})}{\|f(\boldsymbol{x})\|} \right)^T \frac{f(\boldsymbol{x}^c)}{\|f(\boldsymbol{x}^c)\|} - \frac{1}{m} \sum_{i=1}^{m} \left( \frac{f(\boldsymbol{x})}{\|f(\boldsymbol{x})\|} \right)^T \frac{f(\boldsymbol{x}^{(i)})}{\|f(\boldsymbol{x}^{(i)})\|} \quad \text{(H.12)}$$

$$= \left( \frac{f(\boldsymbol{x})}{\|f(\boldsymbol{x})\|} \right)^T \left( \frac{1}{|\mathcal{C}|} \sum_{\boldsymbol{x}_x \in \mathcal{C}} \frac{f(\boldsymbol{x}^c)}{\|f(\boldsymbol{x}^c)\|} \right) - \left( \frac{f(\boldsymbol{x})}{\|f(\boldsymbol{x})\|} \right)^T \left( \frac{1}{m} \sum_{i=1}^{m} \frac{f(\boldsymbol{x}^{(i)})}{\|f(\boldsymbol{x}^{(i)})\|} \right) \quad \text{(H.13)}$$

$$= \left( \frac{f(\boldsymbol{x})}{\|f(\boldsymbol{x})\|} \right)^T \left( \frac{1}{|\mathcal{C}|} \sum_{\boldsymbol{x}^c \in \mathcal{C}} \frac{f(\boldsymbol{x}^c)}{\|f(\boldsymbol{x}^c)\|} - \frac{1}{m} \sum_{i=1}^{m} \frac{f(\boldsymbol{x}^{(i)})}{\|f(\boldsymbol{x}^{(i)})\|} \right). \quad \text{(H.14)}$$

By the definition of $\gamma_{f,\mathcal{C},\mathcal{D}_{foil}}(\boldsymbol{x})$, linear algebra, and the linearity of expectation, we have

$$\gamma_{f,\mathcal{C},\mathcal{D}_{foil}}(\boldsymbol{x}) = \frac{1}{|\mathcal{C}|} \sum_{\boldsymbol{x}^c \in \mathcal{C}} \left( \frac{f(\boldsymbol{x})}{\|f(\boldsymbol{x})\|} \right)^T \frac{f(\boldsymbol{x}^c)}{\|f(\boldsymbol{x}^c)\|} - \mathbb{E}_{\boldsymbol{x}^v \sim \mathcal{D}_{foil}} \left[ \left( \frac{f(\boldsymbol{x})}{\|f(\boldsymbol{x})\|} \right)^T \frac{f(\boldsymbol{x}^v)}{\|f(\boldsymbol{x}^v)\|} \right] \quad \text{(H.15)}$$

$$= \left( \frac{f(\boldsymbol{x})}{\|f(\boldsymbol{x})\|} \right)^T \left( \frac{1}{|\mathcal{C}|} \sum_{\boldsymbol{x}^c \in \mathcal{C}} \frac{f(\boldsymbol{x}^c)}{\|f(\boldsymbol{x}^c)\|} \right) - \left( \frac{f(\boldsymbol{x})}{\|f(\boldsymbol{x})\|} \right)^T \mathbb{E}_{\boldsymbol{x}^v \sim \mathcal{D}_{foil}} \left[ \frac{f(\boldsymbol{x}^v)}{\|\boldsymbol{x}^v\|} \right] \quad \text{(H.16)}$$

$$= \left( \frac{f(\boldsymbol{x})}{\|f(\boldsymbol{x})\|} \right)^T \left( \frac{1}{|\mathcal{C}|} \sum_{\boldsymbol{x}^c \in \mathcal{C}} \frac{f(\boldsymbol{x}^c)}{\|f(\boldsymbol{x}^c)\|} - \mathbb{E}_{\boldsymbol{x}^v \sim \mathcal{D}_{foil}} \left[ \frac{f(\boldsymbol{x}^v)}{\|f(\boldsymbol{x}^v)\|} \right] \right). \quad \text{(H.17)}$$

□

### H.4   PROOF OF **PROPOSITION F.1**

*Proof.* By Mercer's Theorem (Mercer, 1909) (see Ghojogh et al. (2021) for an accessible survey relevant to machine learning), we have

$$\gamma_{f,\mathcal{C},\mathcal{D}_{foil}}(\boldsymbol{x}) = \frac{1}{|\mathcal{C}|} \sum_{\boldsymbol{x}^c \in \mathcal{C}} \psi(f(\boldsymbol{x}))^T \psi(f(\boldsymbol{x}^c)) - \mathbb{E}_{\boldsymbol{x}^v \sim \mathcal{D}_{foil}}[\psi(f(\boldsymbol{x}))^T \psi(f(\boldsymbol{x}^v))] \tag{H.18}$$

$$= \psi(f(\boldsymbol{x}))^T \left( \frac{1}{|\mathcal{C}|} \sum_{\boldsymbol{x}^c \in \mathcal{C}} \psi(f(\boldsymbol{x}^c)) \right) - \psi(f(\boldsymbol{x}))^T \mathbb{E}_{\boldsymbol{x}^v \sim \mathcal{D}_{foil}}[\psi(f(\boldsymbol{x}^v))] \tag{H.19}$$

$$= \psi(f(\boldsymbol{x}))^T \left( \frac{1}{|\mathcal{C}|} \sum_{\boldsymbol{x}^c \in \mathcal{C}} \psi(f(\boldsymbol{x}^c)) - \mathbb{E}_{\boldsymbol{x}^v \sim \mathcal{D}_{foil}}[\psi(f(\boldsymbol{x}^v))] \right), \tag{H.20}$$

Similarly, we have

$$\hat{\gamma}_{f,\mathcal{C},\mathcal{F}}(\boldsymbol{x}) = \frac{1}{|\mathcal{C}|} \sum_{\boldsymbol{x}^c \in \mathcal{C}} \psi(f(\boldsymbol{x}))^T \psi(f(\boldsymbol{x}^c)) - \frac{1}{m} \sum_{i=1}^{m} \psi(f(\boldsymbol{x}))^T \psi(f(\boldsymbol{x}^{(i)})) \tag{H.21}$$

$$= \psi(f(\boldsymbol{x}))^T \left( \frac{1}{|\mathcal{C}|} \sum_{\boldsymbol{x}^c \in \mathcal{C}} \psi(f(\boldsymbol{x}^c)) \right) - \psi(f(\boldsymbol{x}))^T \frac{1}{m} \left( \sum_{i=1}^{m} \psi(f(\boldsymbol{x}^{(i)})) \right) \tag{H.22}$$

$$= \psi(f(\boldsymbol{x}))^T \left( \frac{1}{|\mathcal{C}|} \sum_{\boldsymbol{x}^c \in \mathcal{C}} \psi(f(\boldsymbol{x}^c)) - \frac{1}{m} \sum_{i=1}^{m} \psi(f(\boldsymbol{x}^{(i)})) \right). \tag{H.23}$$

$\square$

### H.5   PROOF OF **PROPOSITION G.1**

*Proof.* Because the sub-corpora are disjoint sets, we have

$$\gamma_{f,\mathcal{C},\mathcal{D}_{foil}}(\boldsymbol{x}) = \frac{1}{|\mathcal{C}|} \sum_{k=1}^{K} \sum_{\boldsymbol{x}^c \in \mathcal{C}_k} s_f(\boldsymbol{x}, \boldsymbol{x}^c) - \mathbb{E}_{\boldsymbol{x}^v \sim \mathcal{D}_{foil}}[s_f(\boldsymbol{x}, \boldsymbol{x}^v)] \tag{H.24}$$

$$= \sum_{k=1}^{K} \frac{|\mathcal{C}_k|}{|C|} \frac{1}{|\mathcal{C}_k|} \sum_{\boldsymbol{x}^c \in \mathcal{C}_k} s_f(\boldsymbol{x}, \boldsymbol{x}^c) - \mathbb{E}_{\boldsymbol{x}^v \sim \mathcal{D}_{foil}}[s_f(\boldsymbol{x}, \boldsymbol{x}^v)] \tag{H.25}$$

$$= \sum_{k=1}^{K} \frac{|\mathcal{C}_k|}{|C|} \frac{1}{|\mathcal{C}_k|} \sum_{\boldsymbol{x}^c \in \mathcal{C}_k} s_f(\boldsymbol{x}, \boldsymbol{x}^c) - \sum_{k=1}^{K} \frac{|\mathcal{C}_k|}{|\mathcal{C}|} \mathbb{E}_{\boldsymbol{x}^v \sim \mathcal{D}_{foil}}[s_f(\boldsymbol{x}, \boldsymbol{x}^v)] \tag{H.26}$$

$$= \sum_{k=1}^{K} \frac{|\mathcal{C}_k|}{|\mathcal{C}|} \left( \frac{1}{|\mathcal{C}_k|} \sum_{\boldsymbol{x}^c \in \mathcal{C}_k} s_f(\boldsymbol{x}, \boldsymbol{x}^c) - \mathbb{E}_{\boldsymbol{x}^v \sim \mathcal{D}_{foil}}[s_f(\boldsymbol{x}, \boldsymbol{x}^v)] \right) \tag{H.27}$$

$$= \sum_{k=1}^{K} \frac{|\mathcal{C}_k|}{|\mathcal{C}|} \gamma_{f,\mathcal{C}_k,\mathcal{D}_{foil}}(\boldsymbol{x}). \tag{H.28}$$

Similarly,

$$\hat{\gamma}_{f,\mathcal{C},\mathcal{F}}(\boldsymbol{x}) = \frac{1}{|\mathcal{C}|} \sum_{k=1}^{K} \sum_{\boldsymbol{x}^c \in \mathcal{C}_k} s_f(\boldsymbol{x}, \boldsymbol{x}^c) - \frac{1}{m} \sum_{i=1}^{m} s_f(\boldsymbol{x}, \boldsymbol{x}^{(i)}) \tag{H.29}$$

$$= \sum_{k=1}^{K} \frac{|\mathcal{C}_k|}{|C|} \frac{1}{|\mathcal{C}_k|} \sum_{\boldsymbol{x}^c \in \mathcal{C}_k} s_f(\boldsymbol{x}, \boldsymbol{x}^c) - \frac{1}{m} \sum_{i=1}^{m} s_f(\boldsymbol{x}, \boldsymbol{x}^{(i)}) \tag{H.30}$$

$$= \sum_{k=1}^{K} \frac{|\mathcal{C}_k|}{|C|} \frac{1}{|\mathcal{C}_k|} \sum_{\boldsymbol{x}^c \in \mathcal{C}_k} s_f(\boldsymbol{x}, \boldsymbol{x}^c) - \sum_{k=1}^{K} \frac{|\mathcal{C}_k|}{|\mathcal{C}|} \left( \frac{1}{m} \sum_{i=1}^{m} s_f(\boldsymbol{x}, \boldsymbol{x}^{(i)}) \right) \tag{H.31}$$

$$= \sum_{k=1}^{K} \frac{|\mathcal{C}_k|}{|\mathcal{C}|} \left( \frac{1}{|\mathcal{C}_k|} \sum_{\boldsymbol{x}^c \in \mathcal{C}_k} s_f(\boldsymbol{x}, \boldsymbol{x}^c) - \frac{1}{m} \sum_{i=1}^{m} s_f(\boldsymbol{x}, \boldsymbol{x}^{(i)}) \right) \tag{H.32}$$

$$= \sum_{k=1}^{K} \frac{|\mathcal{C}_k|}{|\mathcal{C}|} \hat{\gamma}_{f,\mathcal{C}_k,\mathcal{F}}(\boldsymbol{x}). \tag{H.33}$$

$\square$

## I  EXPERIMENT DETAILS

Code is available at `https://github.com/suinleelab/cl-explainability`.

### I.1  DATASETS

**ImageNet.** The ImageNet ILSVRC dataset contains 1.2 million labeled training images and 50,000 labeled validation images over 1,000 object classes (Russakovsky et al., 2015). Since the SimCLR model and its downstream linear classifier are trained and tuned with only the training set, we use the validation set as a held-out set for quantitative evaluation (Section 4.1) and understanding data augmentations (Section 4.2). For computational feasibility, a subset of 10 easily classified classes called ImageNette (including tench, English springer, cassette player, chainsaw, church, French horn, garbage truck, gas pump, golf ball, and parachute) (Howard, 2019) are used for quantitative evaluation.

**CIFAR-10.** The CIFAR-10 dataset consists of 50,000 labeled training images and 10,000 labeled test images of size $32 \times 32$ over 10 classes. The test set is used for quantitative evaluation.

**MURA.** The MURA (MUsculoskeletal RAdiographs) dataset contains 36,808 training radiograph images from 13,457 musculoskeletal studies and 3,197 validation images from 1,199 studies. Each study and its associated images are labeled by radiologists as either normal or abnormal. We further split the official training images into an 80% subset for our model training and a 20% subset for hyperparameter tuning. The official validation set is held out and used for quantitative evaluation.

### I.2  MODELS

**SimCLR.** We obtained the weights of SimCLRv1 (with a ResNet50 backbone) and its downstream linear classifier, pre-trained with ImageNet, from `https://github.com/google-research/simclr` by Chen et al. (2020). The pre-trained SimCLR model and linear classifier were converted to the PyTorch format following the conversion in `https://github.com/tonylins/simclr-converter`, as recommended in the SimCLR authors' official documentation.

**SimSiam.** We trained a SimSiam model (with a ResNet18 backbone) for CIFAR-10 following the training procedure and hyperparameters outlined in Chen & He (2021), using the implementation in `https://github.com/Reza-Safdari/SimSiam-91.9-top1-acc-on-CIFAR10`. The downstream linear classifier was trained with stochastic gradient descent with a learning rate of 30.0 and a batch size of 256 to achieve a top-1 accuracy of 92.06% on the CIFAR-10 test set.

**ResNet classifier.** We trained a ResNet18 classifier for normal vs. abnormal radiograph images with an 80% training subset and tuned hyperparameters with the other 20%. We randomly augmented the data during training by horizontal flipping, vertical flipping, and rotation between $-30°$ and $30°$. The classifier was trained with an Adam optimizer (Kingma & Ba, 2014), with a batch size of 256, a learning rate decay of 0.1 every 10 steps, and a weight decay of 0.001. The initial learning rate was tuned over $\{0.1, 0.01, 0.001, 0.0001\}$ to identify the optimal initial learning rate of 0.001. The trained ResNet18 classifier achieves an accuracy of 80.54% on the held-out official validation set.

### I.3 FEATURE ATTRIBUTION METHODS

**Integrated Gradients.** Integrated Gradients (Sundararajan et al., 2017) with 50 steps for the Riemman approximation of integral were run in this work. The implementation in the Captum package was used (Kokhlikyan et al., 2020). The baseline values correspond to the explicand image with blurring.

**GradientSHAP.** GradientSHAP with 50 random Gaussian noise samples having a standard deviation of 0.2 for the gradient expectation estimation was run. The implementation in the Captum package was used (Kokhlikyan et al., 2020). The baseline values correspond to the explicand image with blurring.

**RISE.** RISE (Petsiuk et al., 2018) with 5000 random masks generated with a masking probability of 0.5 was run. For CIFAR-10, each binary mask before upsampling had size $4 \times 4$. For ImageNet and MURA, each initial binary mask was $7 \times 7$. The baseline values for replacing masked pixels correspond to the explicand image with blurring.

## J CORPUS SIZE SENSITIVITY ANALYSIS

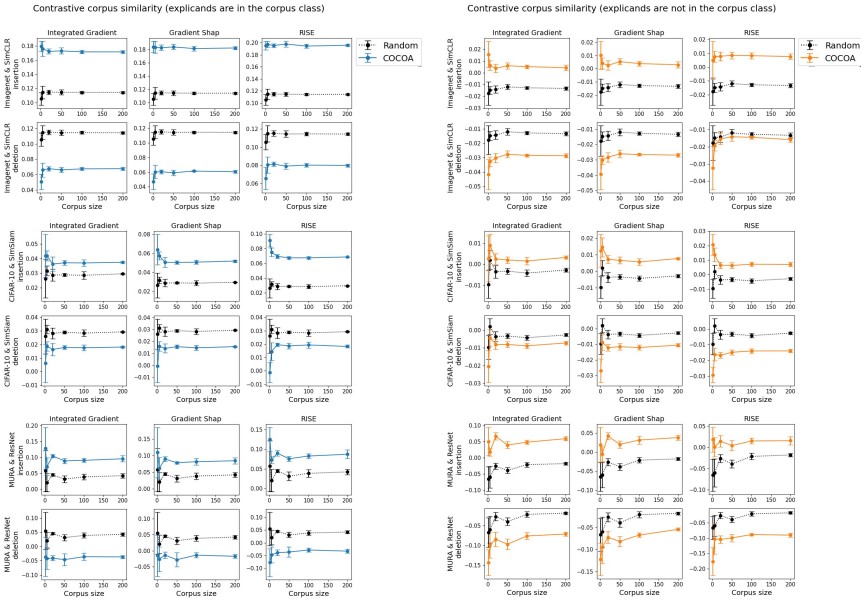

Figure 4: Insertion and deletion metrics of contrastive corpus similarity for COCOA with corpus size $= 1, 5, 20, 50, 100, 200$. The foil size is fixed at 1500. Performance results of random attributions are plotted as benchmarks. Means and 95% confidence intervals across 5 experiment runs are shown.

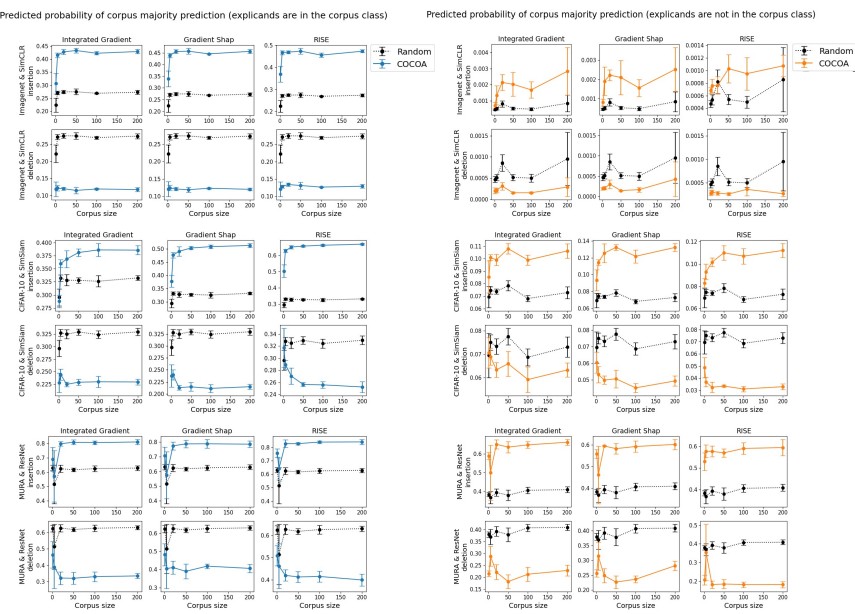

Figure 5: Insertion and deletion metrics of corpus majority probability for COCOA with corpus size $= 1, 5, 20, 50, 100, 200$. The foil size is fixed at $1500$. Performance results of random attributions are plotted as benchmarks. Means and 95% confidence intervals across 5 experiment runs are shown.

## K  FOIL SIZE SENSITIVITY ANALYSIS

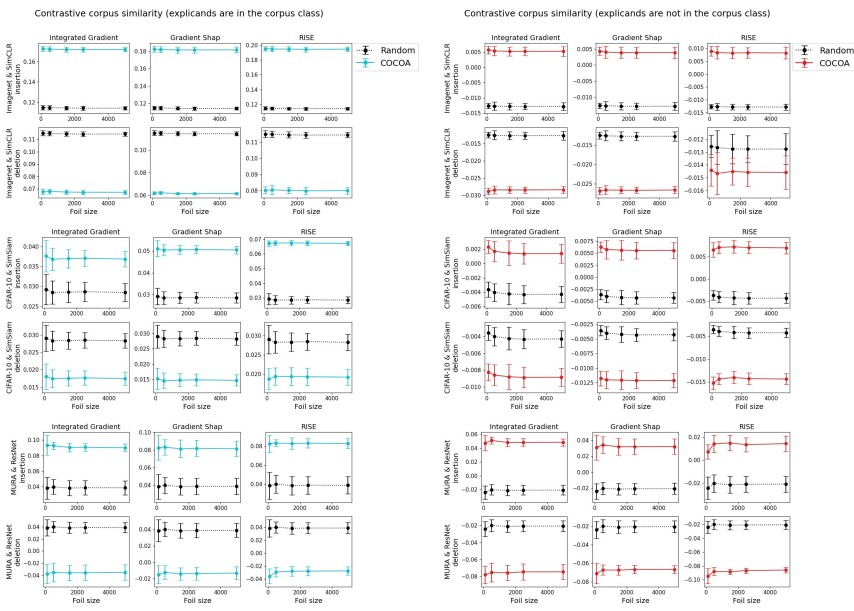

Figure 6: Insertion and deletion metrics of contrastive corpus similarity for COCOA with foil size $= 100, 500, 1500, 2500, 5000$. The corpus size is fixed at $100$. Performance results of random attributions are plotted as benchmarks. Means and 95% confidence intervals across 5 experiment runs are shown.

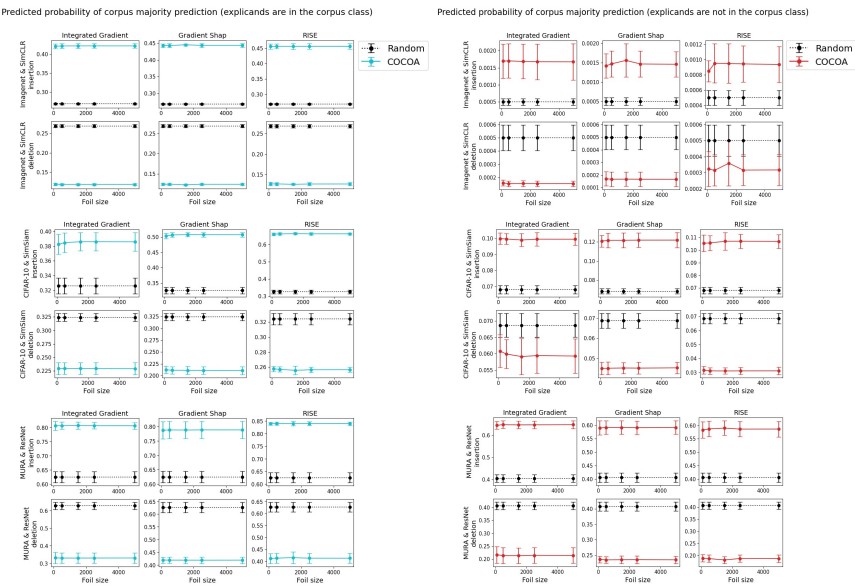

Figure 7: Insertion and deletion metrics of corpus majority probability for COCOA with foil size $= 100, 500, 1500, 2500, 5000$. The corpus size is fixed at 100. Performance results of random attributions are plotted as benchmarks. Means and 95% confidence intervals across 5 experiment runs are shown.

## L  SANITY CHECK RESULTS

Table 3: Insertion and deletion metrics of contrastive corpus similarity when explicands belong to the corpus class. Means (95% confidence intervals) across 5 experiment runs are reported. Higher insertion and lower deletion values indicate better performance, respectively. Each method is a combination of a feature attribution method and an explanation target function (e.g., COCOA under RISE corresponds to feature attributions computed by RISE for the contrastive corpus similarity). Performance results of random attributions (last row) are included as benchmarks.

| Attribution Method | Imagenet & SimCLR | | CIFAR-10 & SimSiam | | MURA & ResNet | |
|---|---|---|---|---|---|---|
| | Insertion (↑) | Deletion (↓) | Insertion (↑) | Deletion (↓) | Insertion (↑) | Deletion (↓) |
| Integrated Gradients | | | | | | |
| Label-Free | 0.154 (0.003) | 0.076 (0.001) | 0.037 (0.002) | 0.021 (0.002) | 0.046 (0.010) | 9.77e-03 ± 1.15e-02 |
| Contrastive Label-Free | 0.161 (0.003) | 0.071 (0.002) | 0.036 (0.003) | 0.020 (0.002) | 0.058 (0.007) | -4.26e-03 ± 1.20e-02 |
| Corpus | 0.157 (0.002) | 0.081 (0.002) | 0.033 (0.002) | 0.020 (0.002) | 0.051 (0.007) | 9.39e-03 ± 1.30e-02 |
| COCOA | **0.172 (0.002)** | **0.067 (0.002)** | **0.037 (0.002)** | **0.018 (0.002)** | **0.091 (0.006)** | **-0.036 (0.013)** |
| Gradient SHAP | | | | | | |
| Label-Free | 0.171 (0.004) | 0.067 (0.001) | 0.048 (0.002) | 0.019 (0.002) | 0.053 (0.013) | 0.017 (0.010) |
| Contrastive Label-Free | 0.173 (0.004) | 0.064 (0.001) | 0.048 (0.002) | 0.019 (0.002) | 0.056 (0.013) | 0.011 (0.007) |
| Corpus | 0.172 (0.004) | 0.071 (0.001) | 0.047 (0.002) | 0.019 (0.002) | 0.064 (0.011) | 9.77e-03 ± 1.26e-02 |
| COCOA | **0.181 (0.004)** | **0.062 (0.001)** | **0.051 (0.002)** | **0.015 (0.002)** | **0.081 (0.010)** | **-0.014 (0.007)** |
| RISE | | | | | | |
| Label-Free (RELAX) | 0.175 (0.003) | 0.091 (0.002) | 0.061 (0.002) | 0.023 (0.002) | 0.049 (0.008) | 0.017 (0.002) |
| Contrastive Label-Free | 0.184 (0.003) | 0.085 (0.002) | 0.062 (0.002) | 0.023 (0.002) | 0.055 (0.006) | -1.08e-03 ± 3.02e-03 |
| Corpus | 0.173 (0.003) | 0.094 (0.003) | 0.059 (0.002) | 0.025 (0.002) | 0.044 (0.010) | 0.026 (0.006) |
| COCOA | **0.195 (0.003)** | **0.080 (0.002)** | **0.068 (0.002)** | **0.019 (0.002)** | **0.083 (0.006)** | **-0.028 (0.008)** |
| Random | 0.115 (0.002) | 0.114 (0.002) | 0.028 (0.002) | 0.029 (0.002) | 0.038 (0.009) | 0.039 (0.009) |

Table 4: Insertion and deletion metrics of contrastive corpus similarity when explicands *do not* belong to the corpus class. Means (95% confidence intervals) across 5 experiment runs are reported. Higher insertion and lower deletion values indicate better performance, respectively. Each method is a combination of a feature attribution method and an explanation target function (e.g., COCOA under RISE corresponds to feature attributions computed by RISE for the contrastive corpus similarity). Performance results of random attributions (last row) are included as benchmarks.

| Attribution Method | Imagenet & SimCLR | | CIFAR-10 & SimSiam | | MURA & ResNet | |
|---|---|---|---|---|---|---|
| | Insertion ($\uparrow$) | Deletion ($\downarrow$) | Insertion ($\uparrow$) | Deletion ($\downarrow$) | Insertion ($\uparrow$) | Deletion ($\downarrow$) |
| Integrated Gradients | | | | | | |
| Label-Free | -0.011 (0.001) | -0.017 (0.001) | -5.34e-03 ± 1.46e-03 | -3.32e-03 ± 1.36e-03 | -0.020 (0.009) | -1.50e-04 ± 8.78e-03 |
| Contrastive Label-Free | -0.011 (0.001) | -0.017 (0.001) | -5.37e-03 ± 1.50e-03 | -3.21e-03 ± 1.36e-03 | -0.035 (0.005) | 0.012 (0.009) |
| Corpus | -4.94e-03 ± 1.38e-03 | -0.020 (0.001) | -2.94e-04 ± 1.78e-03 | -6.37e-03 ± 1.47e-03 | 0.040 (0.008) | -0.062 (0.009) |
| COCOA | **5.24e-03 ± 1.50e-03** | **-0.028 (0.001)** | **1.47e-03 ± 1.74e-03** | **-8.79e-03 ± 1.50e-03** | **0.048 (0.006)** | **-0.076 (0.009)** |
| Gradient SHAP | | | | | | |
| Label-Free | -9.20e-03 ± 1.33e-03 | -0.016 (0.001) | -6.27e-03 ± 1.62e-03 | -3.13e-03 ± 1.57e-03 | -0.037 (0.008) | 3.50e-03 ± 1.03e-02 |
| Contrastive Label-Free | -8.87e-03 ± 1.35e-03 | -0.017 (0.001) | -6.34e-03 ± 1.46e-03 | -3.06e-03 ± 1.56e-03 | -0.035 (0.006) | 3.08e-03 ± 9.38e-03 |
| Corpus | -4.00e-03 ± 1.60e-03 | -0.019 (0.001) | 2.29e-03 ± 1.91e-03 | -8.20e-03 ± 1.39e-03 | 0.018 (0.009) | -0.041 (0.007) |
| COCOA | **3.86e-03 ± 1.76e-03** | **-0.027 (0.001)** | **5.65e-03 ± 2.02e-03** | **-0.012 (0.002)** | **0.032 (0.010)** | **-0.067 (0.005)** |
| RISE | | | | | | |
| Label-Free (RELAX) | -2.87e-03 ± 1.52e-03 | -4.05e-03 ± 1.44e-03 | -6.14e-03 ± 1.31e-03 | -2.18e-03 ± 1.53e-03 | -0.047 (0.006) | -0.024 (0.005) |
| Contrastive Label-Free | -2.87e-03 ± 1.32e-03 | -4.16e-03 ± 1.54e-03 | -6.20e-03 ± 1.38e-03 | -2.16e-03 ± 1.57e-03 | -0.054 (0.006) | -0.017 (0.003) |
| Corpus | -1.08e-03 ± 1.72e-03 | -5.33e-03 ± 1.25e-03 | 1.32e-03 ± 1.73e-03 | -8.70e-03 ± 1.11e-03 | -8.82e-03 ± 6.31e-03 | -0.063 (0.005) |
| COCOA | **8.26e-03 ± 2.17e-03** | **-0.015 (0.001)** | **7.19e-03 ± 1.43e-03** | **-0.014 (0.001)** | **0.015 (0.007)** | **-0.089 (0.003)** |
| Random | -0.013 (0.001) | -0.013 (0.001) | -4.10e-03 ± 1.59e-03 | -4.07e-03 ± 1.46e-03 | -0.021 (0.007) | -0.021 (0.007) |

# M ADDITIONAL QUANTITATIVE EVALUATION RESULTS WITH DOT PRODUCT SIMILARITY

Table 5: Insertion and deletion metrics of contrastive corpus similarity when explicands belong to the corpus class. **All explanation target functions are based on dot product instead of cosine similarity.** Means (95% confidence intervals) across 5 experiment runs are reported. Higher insertion and lower deletion values indicate better performance, respectively. Each method is a combination of a feature attribution method and an explanation target function (e.g., COCOA under RISE corresponds to feature attributions computed by RISE for the contrastive corpus similarity). Performance results of random attributions (last row) are included as benchmarks.

| Attribution Method | Imagenet & SimCLR | | CIFAR-10 & SimSiam | | MURA & ResNet | |
|---|---|---|---|---|---|---|
| | Insertion ($\uparrow$) | Deletion ($\downarrow$) | Insertion ($\uparrow$) | Deletion ($\downarrow$) | Insertion ($\uparrow$) | Deletion ($\downarrow$) |
| Integrated Gradients | | | | | | |
| Label-Free | 0.153 (0.002) | 0.076 (0.002) | 0.031 (0.002) | 0.020 (0.002) | 0.058 (0.008) | -1.40e-03 ± 1.08e-02 |
| Contrastive Label-Free | 0.157 (0.002) | 0.072 (0.002) | 0.032 (0.003) | 0.019 (0.002) | 0.058 (0.007) | -3.08e-03 ± 1.29e-02 |
| Corpus | 0.157 (0.002) | 0.076 (0.002) | 0.031 (0.002) | 0.020 (0.002) | 0.079 (0.008) | -0.018 (0.010) |
| COCOA | **0.168 (0.002)** | **0.068 (0.002)** | **0.036 (0.002)** | **0.017 (0.002)** | **0.091 (0.006)** | **-0.038 (0.013)** |
| Gradient SHAP | | | | | | |
| Label-Free | 0.170 (0.003) | 0.067 (0.001) | 0.045 (0.002) | 0.019 (0.002) | 0.055 (0.013) | 0.013 (0.006) |
| Contrastive Label-Free | 0.174 (0.004) | 0.064 (0.001) | 0.046 (0.002) | 0.018 (0.002) | 0.056 (0.013) | 0.011 (0.006) |
| Corpus | 0.172 (0.003) | 0.068 (0.001) | 0.042 (0.002) | 0.019 (0.002) | 0.073 (0.012) | -2.12e-04 ± 5.80e-03 |
| COCOA | **0.181 (0.003)** | **0.061 (0.001)** | **0.050 (0.002)** | **0.015 (0.002)** | **0.082 (0.010)** | **-0.014 (0.007)** |
| RISE | | | | | | |
| Label-Free | 0.180 (0.003) | 0.087 (0.002) | 0.061 (0.002) | 0.024 (0.002) | 0.053 (0.006) | 7.95e-03 ± 4.51e-03 |
| Contrastive Label-Free | 0.187 (0.004) | 0.083 (0.002) | 0.062 (0.002) | 0.023 (0.002) | 0.056 (0.006) | -3.16e-04 ± 2.35e-03 |
| Corpus | 0.181 (0.003) | 0.088 (0.002) | 0.058 (0.002) | 0.027 (0.002) | 0.062 (0.006) | 4.62e-03 ± 6.23e-03 |
| COCOA | **0.196 (0.004)** | **0.078 (0.002)** | **0.067 (0.002)** | **0.020 (0.002)** | **0.084 (0.006)** | **-0.028 (0.007)** |
| Random | 0.115 (0.002) | 0.114 (0.002) | 0.028 (0.002) | 0.029 (0.002) | 0.038 (0.009) | 0.039 (0.009) |

Table 6: Insertion and deletion metrics of contrastive corpus similarity when explicands *do not* belong to the corpus class. **All explanation target functions are based on dot product instead of cosine similarity.** Means (95% confidence intervals) across 5 experiment runs are reported. Higher insertion and lower deletion values indicate better performance, respectively. Each method is a combination of a feature attribution method and an explanation target function (e.g., COCOA under RISE corresponds to feature attributions computed by RISE for the contrastive corpus similarity). Performance results of random attributions (last row) are included as benchmarks.

| Attribution Method | Imagenet & SimCLR | | CIFAR-10 & SimSiam | | MURA & ResNet | |
|---|---|---|---|---|---|---|
| | Insertion ($\uparrow$) | Deletion ($\downarrow$) | Insertion ($\uparrow$) | Deletion ($\downarrow$) | Insertion ($\uparrow$) | Deletion ($\downarrow$) |
| Integrated Gradients | | | | | | |
| Label-Free | -0.012 (0.001) | -0.015 (0.001) | -4.88e-03 $\pm$ 1.26e-03 | -3.48e-03 $\pm$ 1.48e-03 | -0.039 (0.007) | 0.017 (0.011) |
| Contrastive Label-Free | -0.012 (0.001) | -0.016 (0.001) | -5.11e-03 $\pm$ 1.39e-03 | -3.49e-03 $\pm$ 1.40e-03 | -0.037 (0.005) | 0.015 (0.008) |
| Corpus | -7.35e-03 $\pm$ 1.28e-03 | -0.018 (0.001) | -2.53e-03 $\pm$ 1.46e-03 | -5.20e-03 $\pm$ 1.41e-03 | 0.013 (0.008) | -0.027 (0.007) |
| COCOA | **4.98e-03 $\pm$ 1.51e-03** | **-0.028 (0.001)** | **1.39e-03 $\pm$ 1.90e-03** | **-8.65e-03 $\pm$ 1.54e-03** | **0.051 (0.006)** | **-0.076 (0.007)** |
| Gradient SHAP | | | | | | |
| Label-Free | -9.41e-03 $\pm$ 1.33e-03 | -0.016 (0.001) | -6.11e-03 $\pm$ 1.40e-03 | -3.20e-03 $\pm$ 1.62e-03 | -0.040 (0.005) | 5.79e-03 $\pm$ 1.02e-02 |
| Contrastive Label-Free | -8.91e-03 $\pm$ 1.35e-03 | -0.017 (0.001) | -6.23e-03 $\pm$ 1.37e-03 | -3.07e-03 $\pm$ 1.50e-03 | -0.036 (0.007) | 2.10e-03 $\pm$ 9.37e-03 |
| Corpus | -5.25e-03 $\pm$ 1.44e-03 | -0.019 (0.001) | -1.79e-03 $\pm$ 1.60e-03 | -6.33e-03 $\pm$ 1.31e-03 | 0.019 (0.012) | -0.050 (0.003) |
| COCOA | **3.70e-03 $\pm$ 1.77e-03** | **-0.026 (0.001)** | **5.13e-03 $\pm$ 1.72e-03** | **-0.012 (0.002)** | **0.032 (0.011)** | **-0.068 (0.005)** |
| RISE | | | | | | |
| Label-Free | -3.08e-03 $\pm$ 1.44e-03 | -4.03e-03 $\pm$ 1.43e-03 | -6.13e-03 $\pm$ 1.37e-03 | -2.31e-03 $\pm$ 1.55e-03 | -0.053 (0.006) | -0.019 (0.005) |
| Contrastive Label-Free | -2.86e-03 $\pm$ 1.30e-03 | -4.44e-03 $\pm$ 1.56e-03 | -6.16e-03 $\pm$ 1.33e-03 | -2.22e-03 $\pm$ 1.56e-03 | -0.055 (0.007) | -0.018 (0.002) |
| Corpus | -8.76e-04 $\pm$ 1.64e-03 | -5.55e-03 $\pm$ 1.44e-03 | -1.02e-03 $\pm$ 1.34e-03 | -6.85e-03 $\pm$ 1.14e-03 | -0.018 (0.007) | -0.057 (0.005) |
| COCOA | **8.57e-03 $\pm$ 2.10e-03** | **-0.015 (0.001)** | **7.07e-03 $\pm$ 1.31e-03** | **-0.014 (0.001)** | **0.015 (0.006)** | **-0.088 (0.005)** |
| Random | -0.013 (0.001) | -0.013 (0.001) | -4.10e-03 $\pm$ 1.59e-03 | -4.07e-03 $\pm$ 1.46e-03 | -0.021 (0.007) | -0.021 (0.007) |

Table 7: Insertion and deletion metrics of corpus majority probability when explicands belong to the corpus class. **All explanation target functions are based on dot product instead of cosine similarity.** Means (95% confidence intervals) across 5 experiment runs are reported. Higher insertion and lower deletion values indicate better performance, respectively. Each method is a combination of a feature attribution method and an explanation target function (e.g., COCOA under RISE corresponds to feature attributions computed by RISE for the contrastive corpus similarity). Performance results of random attributions (last row) are included as benchmarks.

| Attribution Method | Imagenet & SimCLR | | CIFAR-10 & SimSiam | | MURA & ResNet | |
|---|---|---|---|---|---|---|
| | Insertion ($\uparrow$) | Deletion ($\downarrow$) | Insertion ($\uparrow$) | Deletion ($\downarrow$) | Insertion ($\uparrow$) | Deletion ($\downarrow$) |
| Integrated Gradients | | | | | | |
| Label-Free | 0.364 (0.004) | 0.125 (0.003) | 0.343 (0.014) | 0.237 (0.012) | 0.696 (0.020) | 0.451 (0.018) |
| Contrastive Label-Free | 0.372 (0.005) | 0.119 (0.003) | 0.354 (0.017) | 0.234 (0.012) | 0.690 (0.022) | 0.452 (0.029) |
| Corpus | 0.384 (0.004) | 0.126 (0.002) | 0.346 (0.012) | 0.240 (0.010) | 0.767 (0.010) | 0.392 (0.018) |
| COCOA | **0.415 (0.006)** | **0.113 (0.002)** | **0.379 (0.012)** | **0.222 (0.009)** | **0.806 (0.012)** | **0.325 (0.031)** |
| Gradient SHAP | | | | | | |
| Label-Free | 0.408 (0.004) | 0.130 (0.002) | 0.475 (0.011) | 0.235 (0.013) | 0.699 (0.040) | 0.515 (0.018) |
| Contrastive Label-Free | 0.413 (0.003) | 0.126 (0.002) | 0.487 (0.011) | 0.229 (0.012) | 0.700 (0.041) | 0.510 (0.019) |
| Corpus | 0.423 (0.004) | 0.130 (0.002) | 0.444 (0.011) | 0.234 (0.008) | 0.762 (0.035) | 0.468 (0.016) |
| COCOA | **0.444 (0.006)** | **0.122 (0.002)** | **0.502 (0.008)** | **0.207 (0.008)** | **0.789 (0.031)** | **0.417 (0.013)** |
| RISE | | | | | | |
| Label-Free | 0.414 (0.007) | 0.145 (0.003) | 0.627 (0.007) | 0.288 (0.004) | 0.724 (0.019) | 0.562 (0.010) |
| Contrastive Label-Free | 0.434 (0.008) | 0.134 (0.004) | 0.633 (0.008) | 0.280 (0.005) | 0.737 (0.021) | 0.533 (0.011) |
| Corpus | 0.417 (0.010) | 0.145 (0.005) | 0.592 (0.005) | 0.315 (0.006) | 0.765 (0.011) | 0.542 (0.019) |
| COCOA | **0.465 (0.009)** | **0.120 (0.004)** | **0.664 (0.007)** | **0.257 (0.006)** | **0.844 (0.006)** | **0.412 (0.025)** |
| Random | 0.269 (0.003) | 0.268 (0.002) | 0.329 (0.013) | 0.329 (0.010) | 0.624 (0.018) | 0.629 (0.018) |

Table 8: Insertion and deletion metrics of corpus majority probability when explicands *do not* belong to the corpus class. **All explanation target functions are based on dot product instead of cosine similarity.** Means (95% confidence intervals) across 5 experiment runs are reported. Higher insertion and lower deletion values indicate better performance, respectively. Each method is a combination of a feature attribution method and an explanation target function (e.g., COCOA under RISE corresponds to feature attributions computed by RISE for the contrastive corpus similarity). Performance results of random attributions (last row) are included as benchmarks.

| Attribution Method | Imagenet & SimCLR | | CIFAR-10 & SimSiam | | MURA & ResNet | |
|---|---|---|---|---|---|---|
| | Insertion (↑) | Deletion (↓) | Insertion (↑) | Deletion (↓) | Insertion (↑) | Deletion (↓) |
| Integrated Gradients | | | | | | |
| Label-Free | 2.74e-04 ± 9.69e-05 | 5.42e-04 ± 1.12e-04 | 0.066 (0.003) | 0.078 (0.004) | 0.352 (0.024) | 0.522 (0.029) |
| Contrastive Label-Free | 2.75e-04 ± 1.00e-04 | 5.00e-04 ± 1.07e-04 | 0.064 (0.004) | 0.078 (0.004) | 0.347 (0.024) | 0.529 (0.023) |
| Corpus | 5.60e-04 ± 2.54e-04 | 4.13e-04 ± 8.00e-05 | 0.079 (0.004) | 0.070 (0.004) | 0.529 (0.018) | 0.381 (0.026) |
| COCOA | **1.58e-03 ± 4.99e-04** | **1.64e-04 ± 2.97e-05** | **0.100 (0.005)** | **0.060 (0.005)** | **0.657 (0.017)** | **0.210 (0.024)** |
| Gradient SHAP | | | | | | |
| Label-Free | 1.88e-04 ± 4.25e-05 | 6.01e-04 ± 1.13e-04 | 0.055 (0.005) | 0.082 (0.006) | 0.343 (0.016) | 0.488 (0.020) |
| Contrastive Label-Free | 2.03e-04 ± 3.73e-05 | 5.30e-04 ± 7.78e-05 | 0.053 (0.004) | 0.082 (0.005) | 0.353 (0.019) | 0.478 (0.019) |
| Corpus | 4.31e-04 ± 1.08e-04 | 4.56e-04 ± 9.36e-05 | 0.081 (0.008) | 0.065 (0.006) | 0.548 (0.036) | 0.303 (0.012) |
| COCOA | **1.43e-03 ± 3.30e-04** | **1.70e-04 ± 6.64e-05** | **0.119 (0.006)** | **0.047 (0.002)** | **0.592 (0.025)** | **0.234 (0.014)** |
| RISE | | | | | | |
| Label-Free | 3.64e-04 ± 8.02e-05 | 5.40e-04 ± 1.11e-04 | 0.040 (0.003) | 0.079 (0.006) | 0.311 (0.022) | 0.450 (0.033) |
| Contrastive Label-Free | 3.23e-04 ± 4.88e-05 | 6.25e-04 ± 1.01e-04 | 0.040 (0.003) | 0.080 (0.005) | 0.303 (0.026) | 0.466 (0.024) |
| Corpus | 5.19e-04 ± 1.06e-04 | 5.69e-04 ± 2.08e-04 | 0.070 (0.004) | 0.056 (0.003) | 0.452 (0.029) | 0.294 (0.020) |
| COCOA | **8.65e-04 ± 1.57e-04** | **3.06e-04 ± 5.43e-05** | **0.105 (0.005)** | **0.032 (0.002)** | **0.593 (0.028)** | **0.177 (0.016)** |
| Random | 4.87e-04 ± 9.50e-05 | 5.03e-04 ± 9.73e-05 | 0.070 (0.003) | 0.070 (0.004) | 0.406 (0.013) | 0.407 (0.016) |

# N  QUANTITATIVE EVALUATION RESULTS WITH MODEL PARAMETER RANDOMIZATION.

It has been shown that feature attribution methods, especially those based on gradients, can be undesirably invariant to model parameter randomization, suggesting that the attributions are not related to the trained models (Adebayo et al., 2018). We performed additional experiments to show that our empirical evaluation results do not suffer from this issue. We generated COCOA scores with randomized models, where the parameters are randomized by sampling from truncated normal distributions (as in Adebayo et al. (2018)), and then evaluated the performance of these COCOA scores. As shown in Tables 9-12, the performance results of COCOA with randomized models are different from and worse than the performance of COCOA with trained models. These results suggest that COCOA performance is indeed sensitive to model parameter randomization and is related to model parameters.

Table 9: COCOA Insertion and deletion metrics of contrastive corpus similarity when explicands belong to the corpus class. Model parameters of randomized models were randomly sampled from truncated normal distributions. Means (95% confidence intervals) across 5 experiment runs are reported. Higher insertion and lower deletion values indicate better performance, respectively. Performance results of random attributions (last row) are included as benchmarks.

| Attribution Method | Imagenet & SimCLR | | CIFAR-10 & SimSiam | | MURA & ResNet | |
|---|---|---|---|---|---|---|
| | Insertion (↑) | Deletion (↓) | Insertion (↑) | Deletion (↓) | Insertion (↑) | Deletion (↓) |
| Integrated Gradients | | | | | | |
| COCOA (trained model) | **0.172 (0.002)** | **0.067 (0.002)** | **0.037 (0.002)** | **0.018 (0.002)** | **0.091 (0.006)** | **-0.036 (0.013)** |
| COCOA (randomized model) | 0.148 (0.003) | 0.147 (0.002) | 0.025 (0.003) | 0.026 (0.003) | 0.040 (0.009) | 0.038 (0.010) |
| Gradient SHAP | | | | | | |
| COCOA (trained model) | **0.181 (0.004)** | **0.062 (0.001)** | **0.051 (0.002)** | **0.015 (0.002)** | **0.081 (0.010)** | **-0.014 (0.007)** |
| COCOA (randomized model) | 0.148 (0.003) | 0.147 (0.002) | 0.026 (0.002) | 0.026 (0.003) | 0.041 (0.008) | 0.039 (0.011) |
| RISE | | | | | | |
| COCOA (trained model) | **0.195 (0.003)** | **0.080 (0.002)** | **0.068 (0.002)** | **0.019 (0.002)** | **0.083 (0.006)** | **-0.028 (0.008)** |
| COCOA (randomized model) | 0.147 (0.004) | 0.149 (0.003) | 0.042 (0.002) | 0.041 (0.002) | 0.033 (0.006) | 0.036 (0.005) |
| Random | 0.115 (0.002) | 0.114 (0.002) | 0.028 (0.002) | 0.029 (0.002) | 0.038 (0.009) | 0.039 (0.009) |

Table 10: COCOA Insertion and deletion metrics of contrastive corpus similarity when explicands *do not* belong to the corpus class. Model parameters of randomized models were randomly sampled from truncated normal distributions. Means (95% confidence intervals) across 5 experiment runs are reported. Higher insertion and lower deletion values indicate better performance, respectively. Performance results of random attributions (last row) are included as benchmarks.

| Attribution Method | Imagenet & SimCLR | | CIFAR-10 & SimSiam | | MURA & ResNet | |
|---|---|---|---|---|---|---|
| | Insertion ($\uparrow$) | Deletion ($\downarrow$) | Insertion ($\uparrow$) | Deletion ($\downarrow$) | Insertion ($\uparrow$) | Deletion ($\downarrow$) |
| Integrated Gradients | | | | | | |
| COCOA (trained model) | **5.24e-03 $\pm$ 1.50e-03** | **-0.028 (0.001)** | **1.47e-03 $\pm$ 1.74e-03** | **-8.79e-03 $\pm$ 1.50e-03** | **0.048 (0.006)** | **-0.076 (0.009)** |
| COCOA (randomized model) | -1.98e-03 $\pm$ 1.42e-03 | -4.53e-03 $\pm$ 1.62e-03 | -4.01e-03 $\pm$ 1.64e-03 | -3.95e-03 $\pm$ 1.70e-03 | -0.010 (0.010) | -0.013 (0.008) |
| Gradient SHAP | | | | | | |
| COCOA (trained model) | **3.86e-03 $\pm$ 1.76e-03** | **-0.027 (0.001)** | **5.65e-03 $\pm$ 2.02e-03** | **-0.012 (0.002)** | **0.032 (0.010)** | **-0.067 (0.005)** |
| COCOA (randomized model) | -1.99e-03 $\pm$ 1.37e-03 | -4.42e-03 $\pm$ 1.75e-03 | -3.91e-03 $\pm$ 1.56e-03 | -3.82e-03 $\pm$ 1.70e-03 | -0.014 (0.009) | -0.017 (0.009) |
| RISE | | | | | | |
| COCOA (trained model) | **8.26e-03 $\pm$ 2.17e-03** | **-0.015 (0.001)** | **7.19e-03 $\pm$ 1.43e-03** | **-0.014 (0.001)** | **0.015 (0.007)** | **-0.089 (0.003)** |
| COCOA (randomized model) | -2.08e-03 $\pm$ 1.43e-03 | -4.42e-03 $\pm$ 1.64e-03 | -3.57e-03 $\pm$ 1.40e-03 | -4.61e-03 $\pm$ 1.61e-03 | -0.040 (0.008) | -0.036 (0.005) |
| Random | -0.013 (0.001) | -0.013 (0.001) | -4.10e-03 $\pm$ 1.59e-03 | -4.07e-03 $\pm$ 1.46e-03 | -0.021 (0.007) | -0.021 (0.007) |

Table 11: COCOA Insertion and deletion metrics of corpus majority probability when explicands belong to the corpus class. Model parameters of randomized models were randomly sampled from truncated normal distributions. Means (95% confidence intervals) across 5 experiment runs are reported. Higher insertion and lower deletion values indicate better performance, respectively. Performance results of random attributions (last row) are included as benchmarks.

| Attribution Method | Imagenet & SimCLR | | CIFAR-10 & SimSiam | | MURA & ResNet | |
|---|---|---|---|---|---|---|
| | Insertion ($\uparrow$) | Deletion ($\downarrow$) | Insertion ($\uparrow$) | Deletion ($\downarrow$) | Insertion ($\uparrow$) | Deletion ($\downarrow$) |
| Integrated Gradients | | | | | | |
| COCOA (trained model) | **0.422 (0.006)** | **0.119 (0.003)** | **0.386 (0.012)** | **0.230 (0.011)** | **0.807 (0.013)** | **0.330 (0.030)** |
| COCOA (randomized model) | 0.321 (0.005) | 0.317 (0.007) | 0.299 (0.011) | 0.303 (0.009) | 0.603 (0.017) | 0.598 (0.019) |
| Gradient SHAP | | | | | | |
| COCOA (trained model) | **0.445 (0.003)** | **0.123 (0.002)** | **0.508 (0.007)** | **0.211 (0.008)** | **0.788 (0.030)** | **0.419 (0.013)** |
| COCOA (randomized model) | 0.321 (0.005) | 0.315 (0.007) | 0.299 (0.008) | 0.305 (0.009) | 0.622 (0.021) | 0.613 (0.019) |
| RISE | | | | | | |
| COCOA (trained model) | **0.456 (0.009)** | **0.126 (0.001)** | **0.663 (0.006)** | **0.256 (0.006)** | **0.840 (0.009)** | **0.415 (0.025)** |
| COCOA (randomized model) | 0.321 (0.008) | 0.322 (0.005) | 0.455 (0.008) | 0.433 (0.015) | 0.649 (0.025) | 0.662 (0.007) |
| Random | 0.269 (0.003) | 0.268 (0.002) | 0.329 (0.013) | 0.329 (0.010) | 0.624 (0.018) | 0.629 (0.018) |

Table 12: COCOA Insertion and deletion metrics of corpus majority probability when explicands *do not* belong to the corpus class. Model parameters of randomized models were randomly sampled from truncated normal distributions. Means (95% confidence intervals) across 5 experiment runs are reported. Higher insertion and lower deletion values indicate better performance, respectively. Performance results of random attributions (last row) are included as benchmarks.

| Attribution Method | Imagenet & SimCLR | | CIFAR-10 & SimSiam | | MURA & ResNet | |
|---|---|---|---|---|---|---|
| | Insertion ($\uparrow$) | Deletion ($\downarrow$) | Insertion ($\uparrow$) | Deletion ($\downarrow$) | Insertion ($\uparrow$) | Deletion ($\downarrow$) |
| Integrated Gradients | | | | | | |
| COCOA (trained model) | **1.69e-03 $\pm$ 5.07e-04** | **1.55e-04 $\pm$ 2.21e-05** | **0.099 (0.004)** | **0.059 (0.005)** | **0.647 (0.017)** | **0.213 (0.030)** |
| COCOA (randomized model) | 5.52e-04 $\pm$ 9.43e-05 | 4.54e-04 $\pm$ 3.29e-05 | 0.072 (0.004) | 0.072 (0.005) | 0.434 (0.018) | 0.424 (0.006) |
| Gradient SHAP | | | | | | |
| COCOA (trained model) | **1.56e-03 $\pm$ 4.32e-04** | **1.67e-04 $\pm$ 5.66e-05** | **0.122 (0.008)** | **0.045 (0.003)** | **0.592 (0.025)** | **0.236 (0.012)** |
| COCOA (randomized model) | 5.38e-04 $\pm$ 9.44e-05 | 4.58e-04 $\pm$ 4.48e-05 | 0.071 (0.004) | 0.073 (0.005) | 0.427 (0.022) | 0.415 (0.016) |
| RISE | | | | | | |
| COCOA (trained model) | **9.51e-04 $\pm$ 2.58e-04** | **3.60e-04 $\pm$ 1.40e-04** | **0.107 (0.007)** | **0.031 (0.002)** | **0.590 (0.027)** | **0.181 (0.014)** |
| COCOA (randomized model) | 5.30e-04 $\pm$ 1.14e-04 | 4.59e-04 $\pm$ 4.66e-05 | 0.070 (0.003) | 0.057 (0.004) | 0.364 (0.035) | 0.375 (0.015) |
| Random | 4.87e-04 $\pm$ 9.50e-05 | 5.03e-04 $\pm$ 9.73e-05 | 0.070 (0.003) | 0.070 (0.004) | 0.406 (0.013) | 0.407 (0.016) |

# O  ADDITIONAL RESULTS FOR UNDERSTANDING DATA AUGMENTATIONS IN SIMCLR

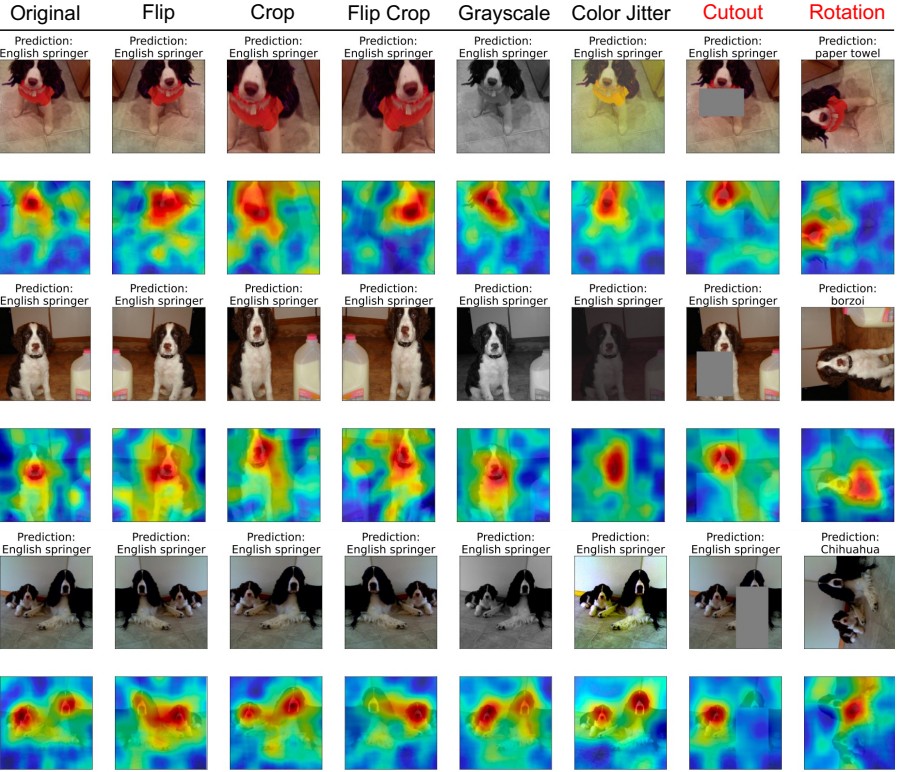

Figure 8: Original version and augmentations of English springer images with their class predictions (top rows), along with the corresponding COCOA attributions (red for higher values and blue for lower values) with each original image as the corpus and random images as the foil (bottom rows). Cutout and rotation are not included in SimCLR training.

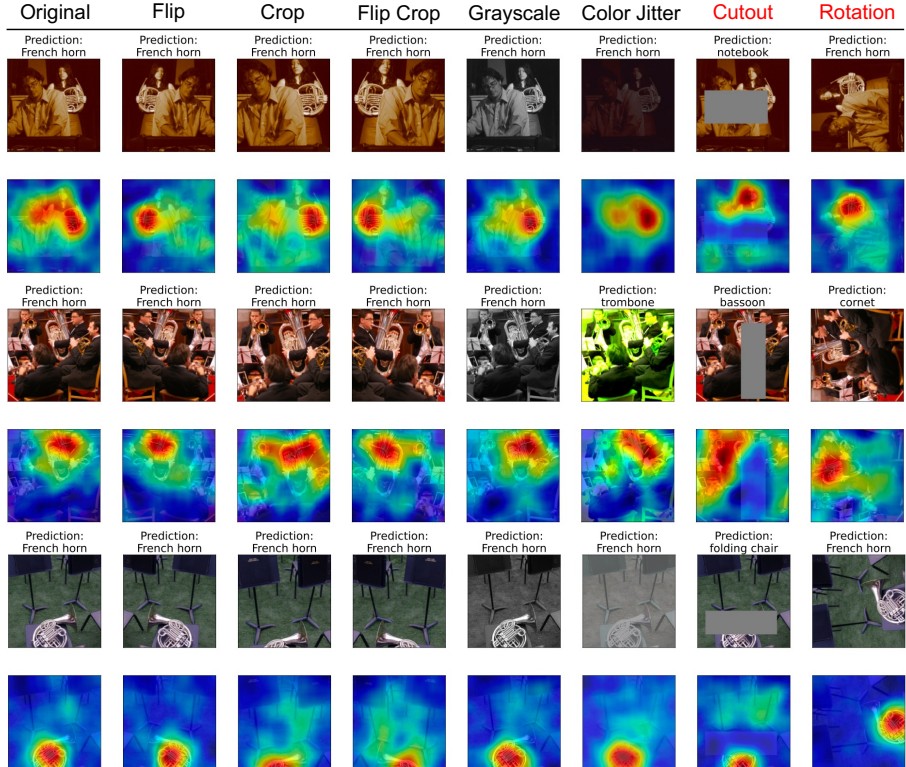

Figure 9: Original version and augmentations of French horn images with their class predictions (top rows), along with the corresponding COCOA attributions (red for higher values and blue for lower values) with each original image as the corpus and random images as the foil (bottom rows). Cutout and rotation are not included in SimCLR training.

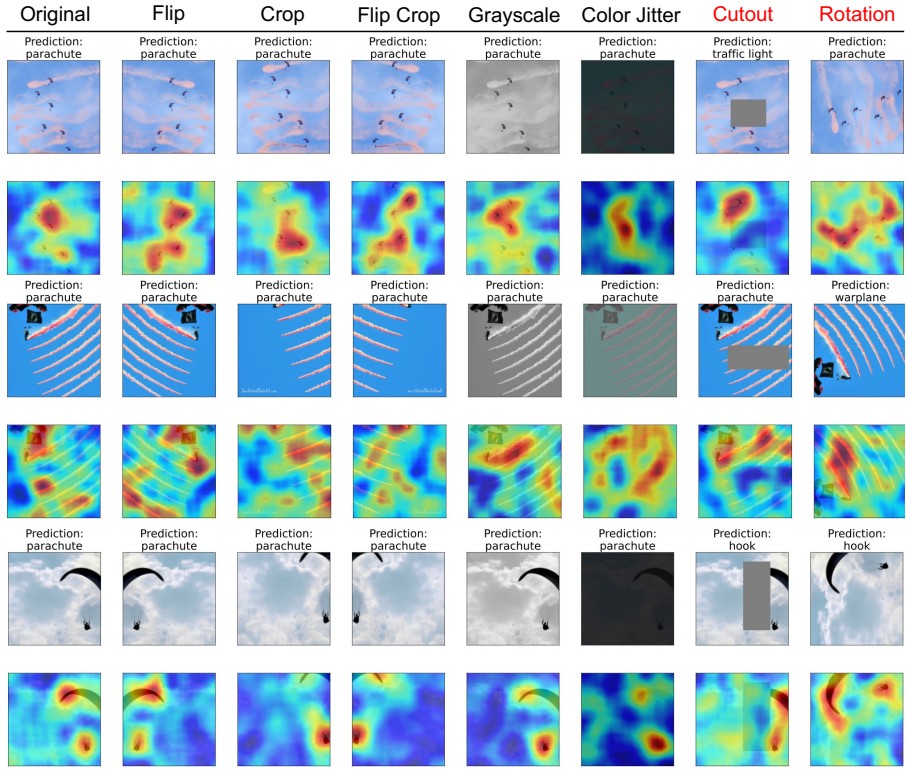

Figure 10: Original version and augmentations of parachute images with their class predictions (top rows), along with the corresponding COCOA attributions (red for higher values and blue for lower values) with each original image as the corpus and random images as the foil (bottom rows). Cutout and rotation are not included in SimCLR training.

