# OpenReview forum: "Contrastive Corpus Attribution for Explaining Representations"
_ICLR.cc/2023/Conference — ICLR 2023 poster_

### Official Review · Reviewer_Y75o · 2022-10-25

**Confidence:** 3
**Correctness:** 3
**Technical Novelty And Significance:** 3
**Empirical Novelty And Significance:** 3
**Recommendation:** 8

**Clarity, Quality, Novelty And Reproducibility:**

The authors appear to have been rigorous, and I didn't check all the proofs in the appendix. The paper is mostly clear, although some of the formulae are quite complex and (arguably) hard to parse quickly. To my knowledge, the work is novel, although I may be unfamiliar with some of the existing literature. The supplementary material includes code, but the documentation (e.g. the README file) is lacking.

**Strength And Weaknesses:**

Strengths

The approach is quite simple, yet effective as demonstrated by the strong results.

The figures clearly help demonstrate the potential impact and applications of the approach.

Weaknesses

To identify the corpus and foil, it is sometimes necessary to have some labeled data.

It is unclear why the results do not include the performance of RELAX.

If the feature representations of the corpus are multimodal (i.e. you get high similarity with some corpus examples, but low similarity with others), the proposed method might arguably assign too low of a score.

**Summary Of The Paper:**

This paper proposes contrastive corpus similarity as a scalar explanation output for unsupervised models. The method relies on a corpus (positive samples) and foil (negative samples or distribution). For a given input, it computes the average cosine similarity between its encoded representation and those of the corpus and subtracts the average similarity with the foil representations. Combined with existing feature attribution methods, the proposed approach can identify relevant input features.

**Summary Of The Review:**

The paper proposes a simple, effective and potentially impactful method to generate contrastive feature attributions for unsupervised models.

---

> ### Author Response · Authors · 2022-11-18
> **Response to Reviewer Y75o**
>
> We thank the reviewer for providing comments and for the positive feedback. Here, we address the comments provided by the reviewer.
>
> > **“To identify the corpus and foil, it is sometimes necessary to have some labeled data.”**
>
> We agree that labels are sometimes, but not always, necessary to apply COCOA. This stems from the fact that contrastive corpus similarity, by definition, does not impose any restrictions on how the corpus and foil should be selected. This freedom of choice allows a wide range of possible use cases, some of which may necessitate some labeled data (*e.g.*, the quantitative evaluation in our paper, which can use as few as 20 labeled corpus samples), while others do not strictly require labeled data (*e.g.*, applications of COCOA demonstrated in Section 4.2 and Section 4.3). Instead of viewing this flexibility as a limitation, we consider it as a desirable characteristic of our method that permits more potential use cases.
> ***
>
> > **“It is unclear why the results do not include the performance of RELAX.”**
>
> RELAX combines representation similarity (defined through the cosine similarity) as its explanation target function and RISE as its choice of feature attribution method. Therefore, in Tables 1, 2, 3, and 4, “Label-Free” (corresponding to representation similarity) under “RISE” actaully corresponds to RELAX. We thank the reviewer for noting this ambiguity and have updated these tables in our revised manuscript to indicate that Label-Free RISE (with cosine similarity) is equivalent to RELAX.
> ***
>
> > **“If the feature representations of the corpus are multimodal (i.e. you get high similarity with some corpus examples, but low similarity with others), the proposed method might arguably assign too low of a score.”**
>
> Suppose the corpus consists of two groups $C_{low}$ and $C_{high}$, where an explicand $x^e$ has low similarity with samples in $C_{low}$ and high similarity with samples in $C_{high}$. Then we have
>
> $\gamma_{f, C, D_{foil}}(x^e) = \frac{|C_{low}|}{|C|} \gamma_{f, C_{low}, D_{foil}}(x^e) + \frac{|C_{high}|}{|C|} \gamma_{f, C_{high}, D_{foil}}(x^e)$
>
> and
>
> $\hat \gamma_{f, C, F}(x^e) = \frac{|C_{low}|}{|C|} \hat \gamma_{f, C_{low}, F}(x^e) + \frac{|C_{high}|}{|C|} \hat \gamma_{f, C_{high}, F}(x^e)$
>
> These equalities mean that the contrastive corpus similarity with respect to the corpus $C$ is a weighted average of the contrastive corpus similarities with respect to $C_{low}$ and $C_{high}$; similarly, the empirical estimator with respect to $C$ is a weighted average of the empirical estimators with respect to $C_{low}$ and $C_{high}$. The generalization of these equations are shown in Appendix G and proven in Appendix H.5 in the updated manuscript.
>
> It is worth noting that the contribution from $C_{low}$ is proportional to its size. Hence, contrastive corpus similarity gives appropriate weight to $C_{low}$ in the scenario brought up by the reviewer.
> ***
>
> > **“The supplementary material includes code, but the documentation (e.g. the README file) is lacking.”**
>
> We have updated our supplementary material to provide better documentation in the README file.

---

> ### Author Response · Authors · 2022-12-05
> **Review Updates**
>
> Thank you again for your positive review! We hope that our response has addressed the questions you had about our work. Please let us know if you have any additional questions. We would be more than happy to address them.

---

### Official Review · Reviewer_9VMz · 2022-10-27

**Confidence:** 3
**Correctness:** 3
**Technical Novelty And Significance:** 3
**Empirical Novelty And Significance:** 3
**Recommendation:** 8

**Clarity, Quality, Novelty And Reproducibility:**

The paper has been mostly straightforward to understand, although some of the equations are really quite complex and difficult and could be challenging to figure out speedily. According to what I understand, the research is innovative, and yet I may not be intimately acquainted with all of the related literature works. There should README file in the supplementary material that could easy for readers to follows.

**Details Of Ethics Concerns:**

No comments

**Strength And Weaknesses:**

Strength
1. The approach is simple, and appears to work well, as demonstrated by the excellent results.
2. The figures hopefully demonstrate how the method might be employed and what sort of impacts it could have.

Limitation
1. More often than not, the author to labeled the data to determine what the corpus and foil are.
2. If the feature representations of the dataset are heterogeneous in nature (high commonality with some corpus examples and low similarity with others), the proposed model might start giving a scoring rate that is excessively limited.

**Summary Of The Paper:**

This study indicates contrastive document similarity as a scalar explanation output for unsupervised methods. A corpora of samples collected and foil are used in the process (negative samples or distribution). It calculates the average cosine similarity between an input's encoded representation and those in the corpus, subtracting the average similarity with foil representations, for the given input. The suggested method can find pertinent input features when used in conjunction with current feature attribution methods.

**Summary Of The Review:**

The article proposes a simple, efficacious, and potentially useful method for generating attribute inferences for unsupervised models that really are different from each other.

---

> ### Author Response · Authors · 2022-11-18
> **Response to Reviewer 9VMz**
>
> We thank the reviewer for the positive feedback, and we are happy to address the comments provided by the reviewer.
>
> > **“More often than not, the author to labeled the data to determine what the corpus and foil are.”**
>
> Our understanding of this comment is that we often need labels to determine corpus and foil sets. We note that whether labels are required depends on the specific use case of COCOA. In our paper, we use labels only for the quantitative evaluation in a more synthetic setting. However, in our more practical use cases showcasing the utility of COCOA (Section 4.2 and 4.3), image labels are not necessarily required.
>
> Specifically, when COCOA is applied to understand an augmented image (Section 4.2), we do not need any labels to identify the original image, which serves as the corpus; and the foil images are randomly drawn from the training set. When COCOA is applied to perform mixed modality object localization (Section 4.3), the corpus caption and foil caption(s) of interest can be freely defined by a user without any image labels, because text and image embeddings in CLIP are connected.
>
> Nevertheless, as in our discussion section, automatic corpus selection can also be a fruitful future direction.
> ***
> > **“If the feature representations of the dataset are heterogeneous in nature (high commonality with some corpus examples and low similarity with others), the proposed model might start giving a scoring rate that is excessively limited.”**
>
> We have included a new theoretical result (Appendix G in the updated manuscript) to show that, when the corpus consists of heterogeneous groups (*i.e.*, sub-corpora), the contrastive corpus similarity is a weighted average of the contrastive sub-corpus similarities. Also, the empirical estimator of contrastive corpus similarity is a weighted average of the estimators of contrastive sub-corpus similarities. The specific equations are:
> $\gamma_{f, C, D_{foil}}(x) = \sum_{k=1}^K \frac{|C_k|}{|C|} \gamma_{f, C_k, D_{foil}}(x) \text{ }(G.1)$
>
> and
>
> $\hat \gamma_{f, C, F}(x) = \sum_{k=1}^K \frac{|C_k|}{|C|} \hat \gamma_{f, C_k, F}(x) \text{ }(G.2)$.
>
> Importantly, the contribution of each sub-corpus is proportional to its size. In this sense, contrastive corpus similarity takes corpus heterogeneity into account by giving higher weights to more commonly occurring sub-corpora.
> ***
>
> > **“There should README file in the supplementary material that could easy for readers to follows.”**
>
> We have updated the supplementary material to provide a better documented README file.

---

> ### Author Response · Authors · 2022-12-05
> **Review Updates**
>
> We would like to thank the reviewer again for the positive review! We hope that our response has addressed the questions in your review. Please let us know if you have any additional questions. We will be sure to address them.

---

### Official Review · Reviewer_chxC · 2022-10-28

**Confidence:** 3
**Clarity, Quality, Novelty And Reproducibility:** 1. Clarity
**Correctness:** 3
**Technical Novelty And Significance:** 3
**Empirical Novelty And Significance:** 3
**Recommendation:** 8

**Strength And Weaknesses:**

### Strengths

1. Writing and notational discipline: The idea is clearly described, and the notations introduced are well-explained. I cannot find any errors in the notations.

2. Simplicity: The idea of using contrastive similarity towards a foil set and away from a corpus set is simple and elegant.

3. Extensive experiments: The experiments are comprehensive, or as extensive as can be expected within one paper. The sanity checking in the Appendix is also appreciated.

### Weaknesses

1. Conceptual Concerns and situating within the broader concerns surrounding XAI (Explainable AI)

The paper early-on cites the paper on "The Mythos of Model Interpretability" by Lipton to effectively state that deep learning methods can be black boxes. However, I believe it fails to contextualize the proposed method within the larger discourse on XAI methods established in that paper. Specifically, the use of attribution maps (or saliency maps or heat maps as alternatively known) has been shown to suffer from several issues. As Barredo Arrieta et al. point out, “there is absolutely no consistency behind what is known as saliency maps, salient masks, heatmaps, neuron activations, attribution, and other approaches alike” [R1, Sec. 5.3]. Also see [R2, R3] for a discussion on the issues with saliency/attribution maps.

The experiments still uses these and other SHAP based methods known to have issues to support their argument. (See [R4] for links to other papers discussing how SHAP can produce inconsistent and unstable explanations).

I'd like to hear the authors' thoughts on why, broadly, their empirical argument which is built upon these methods will not suffer from similar issues. I don't see in the experiments how the proposed method would implicitly be robust to these issues surrounding instability since the method is inherently dependent on having a well-separated concept of foil and corpus samples (or at least that these samples need to already have a human understandable semantic meaning which is subject to biases in the human perception).

One of the core issues of the current state of XAI is how the methods largely depend on biases and interpretation of the observer, and it would be good to establish how this method is not adding to the same issue.


[R1] Explainable artificial intelligence (XAI): Concepts, taxonomies, opportunities and challenges toward responsible AI. - Barredo Arrieta et al.
[R2]  Sanity checks for saliency maps. - Adebayo et al.
[R3] Exploratory Not Explanatory: Counterfactual Analysis of Saliency Maps for Deep Reinforcement Learning - Atrey et al.
[R4] The Disagreement Problem in Explainable Machine Learning: A Practitioner’s Perspective - Krishna et al.

**Summary Of The Paper:**

This paper proposes a method for post-hoc explanations of unsupervised methods. At the heart of this proposed idea is the *contrastive corpus similarity* which employs a corpus set of samples (e.g. abnormal cases in an illustrative task of classifying bone abnormality) and a foil set of samples (e.g. normal cases for the same task). The proposed method uses the notion of contrastive corpus similarity to generate Contrastive Corpus Attributions (COCOA). COCOA works by identifying the features that, when removed, make the representation of an explicand (a sample to be explained) similar to the foil set and dissimilar to the corpus set (hence the term `contrastive' in the name).

The authors evaluate COCOA to establish whether the identified features are indeed related to corpus and foil samples using different models (SimCLR, SimSiam etc.), datasets, and feature attribution methods. They also demonstrate the utility towards understanding image data augmentation and mixed modality object localization.

**Summary Of The Review:**

The proposed idea of generating explanations for unsupervised methods is simple and elegant. Incorporating the notion of contrastive changes to the representation (towards a foil cluster and away from a corpus cluster) in response to changes in the feature space is interesting. The writing is clear and demonstrates notational discipline.

However, my issues with this paper is that it fails in contextualizing how the proposed idea and empirical evaluation do not suffer from well-established issues with the underlying attribution methods used in buildding the empirical evidence in support of the proposed method. (Especially because the paper seems aware of the discourse in the "The Mythos of Model Interpretability" paper). I have added other references to help the authors address this issue. Nevertheless, this prevents me from recommending a higher pre-rebuttal score.

---

> ### Author Response · Authors · 2022-11-18
> **Response to Reviewer chxC (1/2)**
>
> We thank the reviewer for the constructive feedback and thoughtful comments about issues in the field of XAI. Here, we summarize the main points mentioned or cited by the reviewer and address them. Please feel free to post responses for further clarifications or elaborations and let us know if we have misinterpreted any comments.
>
> > **Different feature attribution methods can generate disagreeing results. The proposed idea and empirical evaluation may suffer from this disagreement problem.**
>
> Conceptually, the inconsistency between different feature attribution methods can be an inherent characteristic of the field, in part because some attribution methods are domain specific. For example, RISE is a removal-based method that removes pixels nearby in space. This assumption of spatial locality is appropriate for image data and serves to make the approach computationally efficient (similar assumptions are made by L-Shapley and C-Shapley [1]). However, other attribution methods may be more appropriate for NLP or tabular data. We note that the particular choice of feature attribution method given a specific use case is still an open research area. Fortunately, contrastive corpus similarity is an explanation target function that is compatible with many feature attribution methods. As such, even as new feature attribution methods are proposed, COCOA will still be applicable.
>
> However, we do agree that disagreement between attribution methods is an important issue. This is why we evaluated contrastive corpus similarity with a number of varied attribution methods: a gradient-based method (*i.e.*, Integrated Gradients), a SHAP-related method (*i.e.*, Gradient SHAP), and a removal-based method (*i.e.*, RISE). At the same time, this set of feature attribution methods allows direct comparisons to label-free feature importance (which achieves the best results with Integrated Gradients and Gradient SHAP in the original paper) and RELAX (which uses RISE).
>
> [1] L-Shapley and C-Shapley: Efficient Model Interpretation for Structured Data. - Chen *et al.*
> ***
>
> > **Feature attribution methods, especially those based on gradients, can be invariant to model parameter randomization. The attribution methods used to support the empirical evaluation may suffer from this issue.**
>
> We performed additional experiments to show that our empirical evaluation results do not suffer from this issue. We generated COCOA scores with randomized models, where the parameters are randomized by sampling from truncated normal distributions (as in [R2]), and then evaluated the performance of these COCOA scores. As shown in Tables 9-12 (in Appendix N of the updated manuscript), the performance results of COCOA with randomized models are different from and worse than the performance of COCOA with trained models. This suggests that COCOA is sensitive to model parameter randomization.
> ***

---

> ### Author Response · Authors · 2022-11-18
> **Response to Reviewer chxC (2/2)**
>
> > **The experiments use gradient-based and SHAP-based methods, which are inconsistent and unstable, to support their argument.**
>
> In the citation provided by the reviewer [R4], the inconsistency and instability refer to studies showing that gradient-based methods, SHAP, and LIME can be adversarially manipulated or fairwashed [2, 3].
>
> To the best of our knowledge, existing techniques for manipulating or fairwashing feature attributions focus on the setting of explaining predictions for supervised models [2-5]. Whether these techniques transfer to feature attribution methods for explaining representations remains unexplored. Nevertheless, we seek to provide some conceptual insights about the robustness of COCOA against adversarial manipulation or fairwashing. We observe that existing manipulation techniques require an adversary to have access to the explanation target function or its gradient [2-5]. In the supervised setting, the explanation target function is the model’s predicted probabilities, which only depend on model parameters. In contrast, contrastive corpus similarity (our proposed explanation target function) additionally depends on corpus and foil samples. Hence, to manipulate COCOA, an adversary would need access to not only model parameters, but also the specific corpus and foil that will later be used to explain the model. This makes targeted manipulation of COCOA more difficult. Furthermore, random sampling from the foil distribution and averaging over corpus (foil) samples can be seen as a type of smoothing, which is often used to make feature attributions robust against manipulation [4, 6, 7].
>
> [2] Interpretation of Neural Networks Is Fragile. - Ghorbani and Abid *et al.* [3] Fooling LIME and SHAP: Adversarial Attacks on Post Hoc Explanation Methods. - Slack and Hilgard *et al.* [4] Explanations can be manipulated and geometry is to blame. - Domborwski *et al.* [5] Fairwashing Explanations with Off-Manifold Detergent. - Anders *et al.* [6] Smoothed Geometry for Robust Attribution. - Wang *et al.* [7] Towards robust explanations for deep neural networks. - Dombrowski *et al.*
> ***
>
> > **Semantic meanings of the corpus and the foil are subject to biases in human perception.**
>
> Here, the underlying problem is that, given a set of samples (*i.e.*, images in our paper), different human users may associate them with different semantic meanings. We note that this is a general issue in XAI and machine learning, and there are different approaches to address this issue that are orthogonal to the contributions of our paper. For example, the authors of TCAV collected images related to a semantic concept using a search engine [8] (although search engine results may potentially be biased as well). Alternatively, the authors of ImageNet had multiple users independently label images and considered a label correct only if it received a convincing majority agreement, where the threshold of agreement was determined by an algorithm [9]. Evaluating the efficacy of these approaches and proposing additional ones constitute an important future area of research which may enable us to accurately label and identify semantically uniform samples.
>
> [8] Interpretability Beyond Feature Attribution: Quantitative Testing with Concept Activation Vectors (TCAV). - Kim *et al.* [9] ImageNet: A Large-Scale Hierarchical Image Database. - Deng *et al.*

---

> ### Author Response · Authors · 2022-12-05
> **Review Updates**
>
> We would like to thank the reviewer again for the thoughtful feedback and comments! We hope our response (split into two comments below) has addressed the concerns in your review. Please let us know if you have any additional questions, and we will be sure to address them.

---

### Author Response · Authors · 2022-11-18
**Summary of revision**

We thank the reviewers for their positive feedback and thoughtful comments. We have updated our manuscript in the following ways.

- We have added an additional theoretical result in the Appendix (stated in Section G and proven in Section H.5 of the updated manuscript) to address the scenario when the corpus consists of multiple groups (*i.e.*, sub-corpora). This is to address comments by Reviewer 9VMz and Y75o.

- We have updated Tables 1-4 to indicate that Label-Free RISE (with cosine similarity) is equivalent to RELAX. We thank Reviewer Y75o for noting the ambiguity in our original manuscript.

- We have included new Tables 9-12 (in Appendix N of the updated manuscript) to show that COCOA performance is sensitive to model parameter randomization. This is to address the concern by Reviewer chxC that feature attribution methods may be invariant to parameter randomization.

- We have slightly modified the text in Section 4.1, “(analogous results **using** dot product similarity are in Appendix …)” --> “analogous results **with** dot product similarity are in Appendix …)” to save space.

The supplementary material has also been updated to include a better documented README file as well as code for model parameter randomization.

---

### Decision · Program_Chairs · 2023-01-20

**Decision:**

Accept: poster

**Justification For Why Not Higher Score:**

Had the limitations I listed above been discussed fully and openly in the submitted version, I think we would have pushed the score up.

**Justification For Why Not Lower Score:**

The idea is simple yet elegant, the paper is executed well and the method is shown to be effective. The limitations require some open discussion in the text but are not necessarily motivating much lower scores.

**Metareview: Summary, Strengths And Weaknesses:**

Strengths (supported by all reviewers, myself included):

* clarity
* simple and elegant
* effective
* extensive experiments

Weaknesses

* Method depends on one's ability to assess similarity. This was mentioned by two reviewers, and I agree this is a limitation, but it doesn't seem to be a reason to keep the paper out of ICLR. I recommend the authors to discuss the limitations on the manuscript, as they do in the rebuttal.
* Method requires some labelled data. Also mentioned by two reviewers. I am choosing to see this as a limitation (i.e., it limits applicability) but not a weakness.
* The limitations of saliency maps should be better discussed with a more transparent contextualisation of the method in this landscape. I agree with the reviewer who brought this up and also think it's an important point for discussion in the manuscript. I appreciate that the authors conducted some additional analysis in the rebuttal phase.

I'd like to recommend acceptance, while highly encouraging the authors to be a bit more transparent about the last issue above, and clearly discussing all three points above in the manuscript.

**Note From Pc:**

if the above contains the word "oral" or "spotlight" please see: "oral" presentation means -> notable-top-5% and "spotlight" means -> notable-top-25%. As stated in our emails, we are disassociating presentation type from AC recommendations

**Summary Of Ac-Reviewer Meeting:**

There was no need for such a meeting.